# Feeding state-dependent neuropeptidergic modulation of reciprocally interconnected inhibitory neurons biases sensorimotor decisions in *Drosophila*

Eloïse de Tredern[1,5], Dylan Manceau[1,5], Alexandre Blanc [2,3], Abhijit Parameswaran[1], Panagiotis Sakagiannis [4], Chloe Barre[2,3], Victoria Sus[1], Francesca Viscido [1], Perla Akiki[1], Md Amit Hasan[1], Sandra Autran[1], François Laurent [2,3], Martin Paul Nawrot [4], Jean-Baptiste Masson [2,3] & Tihana Jovanic [1] ✉

An animal's feeding state changes its behavioral priorities and thus influences even nonfeeding-related decisions. How the feeding state information is transmitted to nonfeeding-related circuits and what circuit mechanisms are involved in biasing nonfeeding-related decisions remain open questions. By combining calcium imaging, neuronal manipulations, behavioral analysis and computational modeling, we determined that the competition between different aversive responses to mechanical cues is biased by changes in the feeding state. We found that this effect is achieved by the differential modulation of two different types of reciprocally connected inhibitory neurons promoting opposing actions. This modulation results in a more frequent active type of response and, less frequently, a protective type of response if larvae are fed sugar than when they are fed a balanced diet. Information about the internal state is conveyed to inhibitory neurons through homologs of the vertebrate neuropeptide Y, which is known to be involved in regulating feeding behavior.

Physiological states such as hunger and thirst are powerful regulators of behavior across the animal kingdom because strong homeostatic drives are critical for survival[1–7]. For example, across model systems, food deprivation has been shown to modulate responsiveness to stimuli by influencing sensory neurons and sensory pathways[8–11], suggesting that food deprivation can alter the perceived value of a stimulus, which in turn affects behavioral decisions. Various studies have implicated changes in central processing[12–15], which lead to changes in behavioral decisions in hungry animals. However, the detailed neural circuit mechanisms of this state-dependent flexibility of behaviors remain largely unknown.

Internal drives (e.g., hunger and thirst) need to be balanced by environmental demands such as the need to avoid dangers. Avoiding danger is a critical instinctive behavior that must be balanced with finding and consuming food to ensure survival. Avoidance behaviors tend to be robust, which makes them excellent systems for studying the neural bases of behavior[16–19]; however, they also need to be flexible for animals to adapt their behavioral strategies to different contexts

[1]Université Paris-Saclay, CNRS, Institut des neurosciences Paris-Saclay, Saclay, France. [2]Institut Pasteur, Université Paris Cité, CNRS UMR 3571, Decision and Bayesian Computation, Paris, France. [3]Epiméthée, INRIA, Paris, France. [4]Computational Systems Neuroscience, Institute of Zoology, University of Cologne, Cologne, Germany. [5]These authors contributed equally: Eloïse de Tredern, Dylan Manceau. ✉e-mail: tihana.jovanic@cnrs.fr

and according to different internal states[17,20–22]. Feeding states and contexts can, for example, influence both the tolerance to the level of threat and action selection during threat avoidance[16–18,23].

At the neural circuit level, such behavioral flexibility is thought to be implemented by neuromodulation (modulation of existing synaptic connections by neuropeptides, for instance), which could bias the outcome of competition between diverse behaviors. The outcome could differ depending on the neuropeptide released[24–30]. Alternatively, information rerouting (using alternative circuit pathways)[12,13,31], where information is processed differently depending on the context or state, could alter the behavioral choice in a context-dependent manner. The types of circuit motifs underlying competitive selection must allow for such flexible processing of information. Reciprocal inhibition of inhibition has been proposed to be a motif that could confer the circuits the property to be tuned to contextual/state information and thus implement flexible competitive selection[32–35]. However, the detailed mechanisms and implications in the case of state-dependent flexible selection have not been experimentally demonstrated. In addition, the neural circuit mechanisms underlying the feeding state-dependent modulation of behavior, the neuromodulators involved, and their mechanism of action on specific circuits, especially those that pertain directly to non-feeding or nonwater-seeking behavior, are not well understood.

This paper addresses the understudied aspect of the influence of physiological drive on competitive selection during avoidance behaviors and investigates the neural circuit mechanisms involved in a powerful model organism for neural circuit analysis: the *Drosophila* larva[36]. *Drosophila* larvae are ideally suited for combining comprehensive, synaptic-resolution circuit mapping in electron microscopy (EM) across the nervous system[32,37–39] with the targeted manipulation of uniquely identified circuit motifs at the individual neuron level, enabling the establishment of causal relationships between circuit structure and function in a brain-wide manner. In addition, evolutionarily conserved neuropeptidergic and hormonal pathways in *Drosophila* have been shown to regulate diverse behaviors[11,18,27,40,41].

Previous studies have described the larval avoidance response to a mechanical stimulus (air puff) and detailed the neural circuit underlying the competition between Hunching, a protective or startle-like type of behavior, and Head Casting, an active exploratory type of behavior that can lead to escape[32,42–46]. We identified circuit motifs underlying competitive interactions between behavioral actions (reciprocal inhibition of inhibition) and sequence transitions (lateral and feedback disinhibition) between the two behaviors. These types of motifs based on disinhibition would allow for flexible behavioral selection in a context/state-dependent manner. By combining calcium imaging and neuronal and neuropeptide manipulation at the single-cell level with automated tracking, behavioral classification, and computational modeling, this work shows that larval responses to air puff are biased toward less protective actions and toward more active, exploratory actions upon changes in their feeding conditions. We determine that this bias is due to the differential modulation of two reciprocally connected inhibitory neurons that drive competing behaviors (Hunching and Head Casting): the activity of the neuron that promotes protective hunching is decreased, and the activity of the neuron that inhibits Hunching is increased. We also show that the modulation at the level of reciprocally interconnected inhibitory neurons results in a bias at the level of the network output toward a state that will lead to Head Casting at the expense of Hunching upon feeding on sucrose. Finally, we determine that NPF and sNPF modulate the activity of reciprocally interconnected inhibitory neurons that bias the behavioral output in a feeding state-dependent manner.

## Results

### Changes in feeding conditions for short periods affect larval feeding and locomotion

We sought to establish a short-term food-deprivation protocol to study the effects of diet on behavioral decisions as a result of cognitive control and the prioritization of needs and motivations rather than long-term physiological changes that affect circuit properties. We determined the shortest duration of food deprivation that was sufficient to induce quantifiable changes in behavior. Depriving larvae of food completely (by placing them on water-soaked filter paper) or feeding them 20% sucrose only (and therefore depriving them of proteins and increasing their sugar intake) for 90 mins was sufficient to alter larval locomotion. Compared with normally fed larvae, starved larvae and larvae that fed on sucrose moved at a greater speed and spent more time crawling, eventually dispersing faster in less curved trajectories (Fig. 1a–d and Supplementary Fig. 1a–h). These changes in locomotion are consistent with increased exploration and are likely due to an increased drive to find nutrients caused by deprivation. We monitored locomotion in these larvae upon refeeding them for 15 minutes to determine whether the 90 min feeding deprivation protocol induced changes in behavior that were reversible. Upon refeeding, the starved larvae were slower, and their locomotor phenotype was similar to that of the larvae that were constantly fed (Supplementary Fig. 1i–l).

The consumption rates of different foods were quantified in larvae under different feeding conditions to determine whether the larval need for nutrients was affected by the feeding protocols to which they were subjected (Fig. 1e–j). Compared with both normally fed larvae and larvae that were fed sucrose only, starved larvae significantly increased their intake of standard *Drosophila* food and yeast (rich in amino acids) (Fig. 1e, f), which is consistent with a deficit in nutrients in starved larvae. Larvae fed sucrose only significantly increased their intake of yeast, which is consistent with a deficit in amino acids (Fig. 1f). These larvae, however, did not increase their intake of standard *Drosophila* food, suggesting that the increased sugar consumption might suppress their intake of carbohydrate-rich food. To verify if larvae experience osmotic stress upon sucrose feeding, water consumption was quantified in the different states. Compared with both normally fed and starved larvae, the water consumption of larvae fed 20% sucrose was significantly increased (Fig. 1g), likely due to changes in extracellular osmolality as a result of high sugar intake[6]. Larvae fed standard food supplemented with up to 20% sucrose, on the other hand, did not increase their yeast consumption compared with that of normally fed larvae (Fig. 1h), suggesting that the increase in yeast consumption in only sucrose-fed larvae was due to the lack of protein. To test whether larvae fed on sucrose were repulsed by sugar, a choice assay was performed: larvae were added to the middle of an agar plate where half of the plate was covered with agar and the other half was covered with agar mixed with sucrose. The larvae were then monitored for 15 min (Fig. 1i). After 15 min, more normally fed and starved larvae were found on the agar + sucrose half of the arena, with a preference index increasing over time, whereas larvae fed sucrose showed a decrease in preference for agar supplemented with sucrose. Compared with normally fed and starved larvae, larvae fed only 20% sucrose for 90 min presented decreased sucrose intake (Fig. 1j). Gustatory neurons inhibition (Gr43a-Gal4> TNT) decreases sucrose intake in fed larvae and starved larvae while the consumption of sucrose fed larvae remains the same (Fig. 1j). The sucralose intake of sucrose fed larvae intake was similar to that of normally fed and starved larvae (Fig. 1k), suggesting that their avoidance of sugar was mediated by energy-sensing pathways and not taste. The hemolymph glucose levels were quantified under different feeding conditions to determine whether the 90-minute food-deprivation or sucrose diet protocols caused changes in glucose levels. In larvae that were fed only 20% sucrose, the glucose level was significantly increased compared with

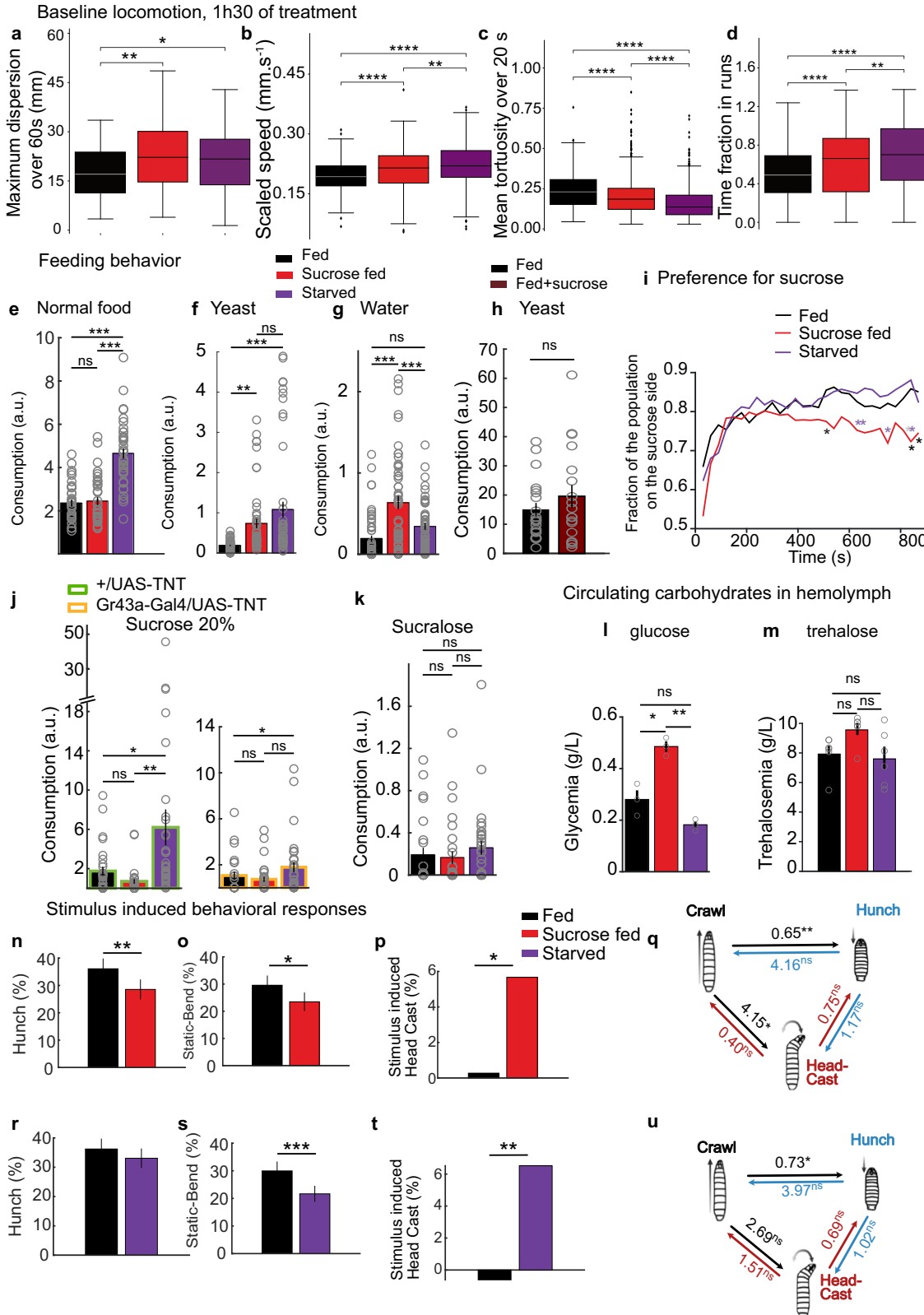

that in both normally fed and starved larvae (Fig. 1l, m). This high glucose level suggests that the consumption of carbohydrates by larvae fed sucrose is suppressed due to high circulating levels of glucose. We therefore tested whether rehydration would reduce the increase in exploration and locomotion observed in larvae fed sucrose. Indeed, after larvae fed sucrose only were placed on water for 15 minutes, their locomotion was similar to that of normally fed larvae (Supplementary

Fig. 1m–p). Furthermore, larvae that fed on sucralose with a similar osmolarity to that of 20% sucrose also increased their exploration and locomotion (Supplementary Fig. 1q–t).

Taken together, these results suggest that depriving larvae of nutrients for 90 min is sufficient to alter feeding and locomotion and that these changes are due to a lack of nutrients in starved larvae and could be due to both protein hunger and thirst in larvae fed sucrose.

**Fig. 1 | Changes in physiology and behavior of food-deprived larvae.**
**a–d** Analysis of larval locomotion. Box plots display the median, interquartile range (IQR), and whiskers up to 1.5 × IQR. Outliers are plotted individually. **e–h, j, k** Larval feeding on different substrates was quantified, in control animals fed ad libitum, in animals fed on sucrose, or subjected to starvation during 90 min. Data are presented as mean values +/− SEM. **e** Starved animals increased their feeding on a standard food medium as compared to fed animals. **f** Both animals fed on sucrose only and starved animals increase their yeast feeding as compared to normally fed larvae. **g** While starved animals consume a similar amount of water as fed ones, animals fed on sucrose only double their water consumption (**h**) Consumption of yeast was similar between larvae fed on normal food and those fed on food with 20% sucrose. **i** Place preference assay for sucrose. Larvae fed on 20% sucrose exhibit a decreased sucrose preference compared to normally fed and starved larvae. **j** Sucrose intake in larvae in different feeding states upon gustatory neurons inhibition (Gr43a-Gal4 > TNT) and in intact larvae. **k** Consumption of sucralose was

similar among conditions. **l–m** Concentration of glucose and trehalose in larval hemolymph. **n–u** Manipulating the feeding state modulates behavioral responses to mechanosensory stimuli. Behavior in response to an air-puff for larvae fed on sucrose only (**n–q**) and starved larvae (**r–u**). **n–p, r–t** Hunch and Static-bend are presented as population probability during the first five seconds upon stimulus onset and 95 % confidence interval, Head Cast as post-stimulus probability corrected by baseline probability. **q, u** Behavioral transitions over the first two seconds of stimulation in sucrose fed larvae (**q**) or starved (**u**) compared to fed larvae (Statistics: **a–d** two-sided Mann-Whitney test with Bonferroni correction; (**e–g, k–m**) one-way ANOVA with Tukey post-hoc test (two-sided); (**j**) one-sided Mann-Whitney test with Bonferroni correction; (**h**) two-tailed T-test. (**i, n, o, r, s** Chi-square (one-sided) test. **p, t** Numerical simulation test; **q, u** Maximum likelihood test (one-sided, chi-square approximation); ****$p < 0.0001$, ***$p < 0.001$, **$p < 0.01$, *$p < 0.05$). See also Supplementary Fig. 1.The source data and $p$-values are provided in Source Data 1–5.

---

The observed increase in locomotion in larvae deprived of nutrients would thus likely result from an increase in the motivational drive to find the missing nutrients.

### Changes in physiological states affect sensorimotor decisions in response to an air puff

We monitored the sensorimotor decisions of *Drosophila* larva in response to an aversive mechanical stimulus, the air puff, to determine whether starvation and a sucrose-only diet can affect nonfeeding-related behaviors. In response to an air puff, larvae perform probabilistic sequences of five mutually exclusive actions that we have characterized in detail in the past[32,47]: Hunch, Bend, Stop, Back-up, and Crawl. We have also identified circuit motifs and characterized the neural circuit mechanisms underlying the competitive interactions between the two most prominent actions (i.e., the Hunch and the Bend) that occur in response to an air puff[32]. The model in that study predicted that the different activation levels of inhibitory neurons determine which actions will take place: the Hunch, Bend, or the Hunch-Bend sequence.

With characterized neurons and synaptic connectivity between the neurons, as well as the availability of driver lines that label the neurons of interest, this circuit provides an excellent system to study whether the activity of the neurons is modulated by changes in an animal's physiological state.

We compared larval behavioral responses to an air puff after subjecting them to the different feeding protocols (as described in the first section of the Results) to determine whether the feeding state modulates larval behavior in response to the mechanosensory stimulus and, by extension, whether sensorimotor decision-making in response to an air puff is an adequate behavioral paradigm to study the effects of food deprivation on neural circuit activity. For this purpose, we used automated tracking[48] to monitor larval behaviors in response to an air puff and updated the supervised machine-learning-based classification method developed in our previous work[42,47] to compute the probabilities of the different actions that occur in response to an air puff. The new classifiers were trained on larvae fed different diets, thus taking into account a broader range of behavioral dynamics and different phenotypes where common actions, e.g., such as Hunch, may have different features, e.g., they can exhibit slower or faster head retraction than Hunches detected previously. We also separated the large Bend behavioral category into two types of bending behavior: Static Bends, a form of protective action where the larva responds to the stimulus by being immobilized in a curved position for a period of time, and exploratory active Head Casts, where the head of the larva moves from one side to the other side, which can lead to escape[42].

Our analysis showed that larvae fed on sucrose Hunch less (and perform fewer Static Bends) and Head Cast more (Fig. 1n–q). Similarly, starved larvae performed fewer Static Bends and more Head Casts (Fig. 1r–u). Increasing the duration of sucrose feeding and starvation to

5 h resulted in similar phenotypes (Supplementary Fig. 1u–z, Source data 3), suggesting that the phenotypes persist over a range of durations of food deprivation.

Using our machine learning-based classification to monitor internal state-induced behavioral changes revealed that changing feeding conditions for short periods of time alters sensorimotor decisions in response to an aversive mechanical cue and results in fewer protective types of actions and more actions consistent with active exploration and escape.

### The feeding state does not affect the response of chordotonal sensory neurons to a mechanical stimulus

As a method to test whether changes in feeding conditions alter the perceived value of a stimulus, which in turn affects behavioral decisions, calcium responses in sensory neurons that sense air puff and chordotonal sensory neurons[32,44,47,48] were monitored in response to a moderately strong mechanical stimulus under three different feeding conditions: normally fed, fed sucrose and starved (Fig. 2a–d and Supplementary Fig. 2). We found no significant differences in chordotonal calcium responses among the different feeding states (Fig. 2b–d, Supplementary Fig. 2a–c), although the responses were slightly higher in starved larvae than in larvae fed standard food or sucrose only. GFP was genetically expressed in the neurons, and its expression level was quantified by comparing the fluorescence in larvae exposed to all three feeding conditions to ensure that the changes in feeding conditions did not influence protein expression and, by extension, GCaMP expression levels. No differences in GFP fluorescence were detected among the three feeding states (Supplementary Fig. 2e–h). The results, therefore, suggest that chordotonal responses are not significantly affected by starvation or sucrose feeding and that the behavioral changes observed under different feeding conditions are not due to changes in stimulus sensitivity at the level of sensory neurons. We further confirmed this result by optogenetically activating chordotonal sensory neurons in larvae in all three feeding states using a driver that labels all eight subtypes of chordotonal sensory neurons: R61D08. We found that although the activation of chordotonal sensory neurons in all three feeding states was the same due to optogenetic stimulation, larvae performed fewer Hunches and more Head Casts when they were fed only sucrose than when they were fed normally (Fig. 2e, f). Similarly, the modulation of behavior when they were starved could be observed upon chordotonal optogenetic activation (Supplementary Fig. 2i, j). Taken together, these results show that feeding state-dependent modulation could target neurons downstream of chordotonal sensory neurons.

### Feeding states modulate the output of the circuit for the choice between Hunch and Head Cast

To determine whether the changes in the feeding state influence the output of the network for the selection between the Hunch and the

**a**  Structure of the mechano-sensory circuit

● mechanosensory neurons
● feedback inhibitory neurons
● feedforward inhibitory neurons
● projection neurons

**b**  Calcium imaging in chordotonal mechanosensory neurons

Behavioral response to optogenetic activation of chordotonal mechanosensory neurons

**Fig. 2 | Feeding state dependent bias in sensorimotor responses does not come from the modulation of sensory neurons. a** Organization of the circuit. **b–d** Calcium responses to mechanical stimulations (5 V) in chordotonal neurons larvae fed on sucrose only and on standard food (R61D08-Gal4/UAS-GCaMP6s). **b** Calcium responses of chordotonal neuron projections in the VNC from different individuals fed on different diets. **c** calcium response averaged during the stimulus. The white line represents the mean, the white dot represents the median, colored dots with white edges represent individual data points. Stimulus-induced activity of chordotonal mechanosensory neurons is unchanged in animals fed on sucrose as compared to larvae fed on standard food. **d** mean calcium trace of chordotonal neurons over time +/- SEM. The green dashed line corresponds to stimulus duration. **e, f** Optogenetic activation of cho in larvae fed only on sucrose. Red light (0.3 mW/cm²) was used for CsChrimson activation. **e** Hunch probability is computed during the first 2 seconds from stimulus onset. **f** Bend is the mean probability during the first 10 s from stimulus onset, corrected by the baseline recording prior to the stimulus. (Hunch presented as population probability and 95 % confidence interval, Head Cast as post-stimulus probability corrected by baseline probability). (Statistics: **c** two-tailed T-test; **e** Chi-square (one-sided) test; **f** Numerical simulation test; ***$p < 0.001$, **$p < 0.01$, *$p < 0.05$). See also Supplementary Fig. 2. The source data and $p$-values are provided in Source Data 2, 5, and 6.

Bend, as characterized in previous work[32], calcium responses of projection neurons Basin-1 and Basin-2 were monitored at different intensities of mechanical stimulation in the three feeding states (Fig. 3a–g and Supplementary Fig. 3). Basin-1 and Basin-2 are differentially involved in Hunching and Bending, as shown in Jovanic et al., 2016[32]: Basin-1 is required for both Hunching and Bending, whereas Basin-2 promotes Bending and inhibits Hunching. The responses of Basin-2 neurons can thus be used as a readout of the Bend state: if Basin-2 is ON, the larva will Bend, and if it is OFF, the larva will Hunch. To confirm that the circuit output and specific Basin-2 function can be mapped on Head Cast and Hunch, responses to a mechanical stimulus in larvae where Basin-2 was silenced with TNT using a Basin-2-specific driver (SS00739) were analyzed using the current classification method, where Bend was separated into Head Cast and Static Bend. We found that Basin-2 is indeed required for Head Cast and inhibited Hunch (Supplementary Fig. 2k–m). Hereafter, we use Hunch and Head Cast to describe the behaviors controlled by the characterized circuit.

We found that for most stimulus intensities, the Basin-1 responses remain only mildly affected by the sucrose-only diet (Fig. 3a–c and Supplementary Fig. 3g). Only for the very weak stimulus intensity did we observe a decrease in the Basin-1 response (Supplementary Fig. 3g). Under starvation conditions, the effect on Basin-1 neurons was similar to that under sucrose-only conditions (Supplementary Fig. 3a–c, h). The activity of Basin-2, on the other hand, was moderately increased in larvae fed only sucrose at all intensities except at the lowest stimulus intensity (Fig. 3d–g and Supplementary Fig. 3i), where no difference was observed between Basin-2 responses under different feeding conditions (Supplementary Fig. 3i, j). However, the activity of Basin-2 in starved larvae was slightly (but not significantly) decreased compared with normally fed larvae at lower intensities of stimulation, whereas it was increased at higher intensities of stimulation (Supplementary Fig. 3d–f, i, j).

We then compared the distributions of Basin-2 responses in larvae fed either standard food or only sucrose to different intensities of

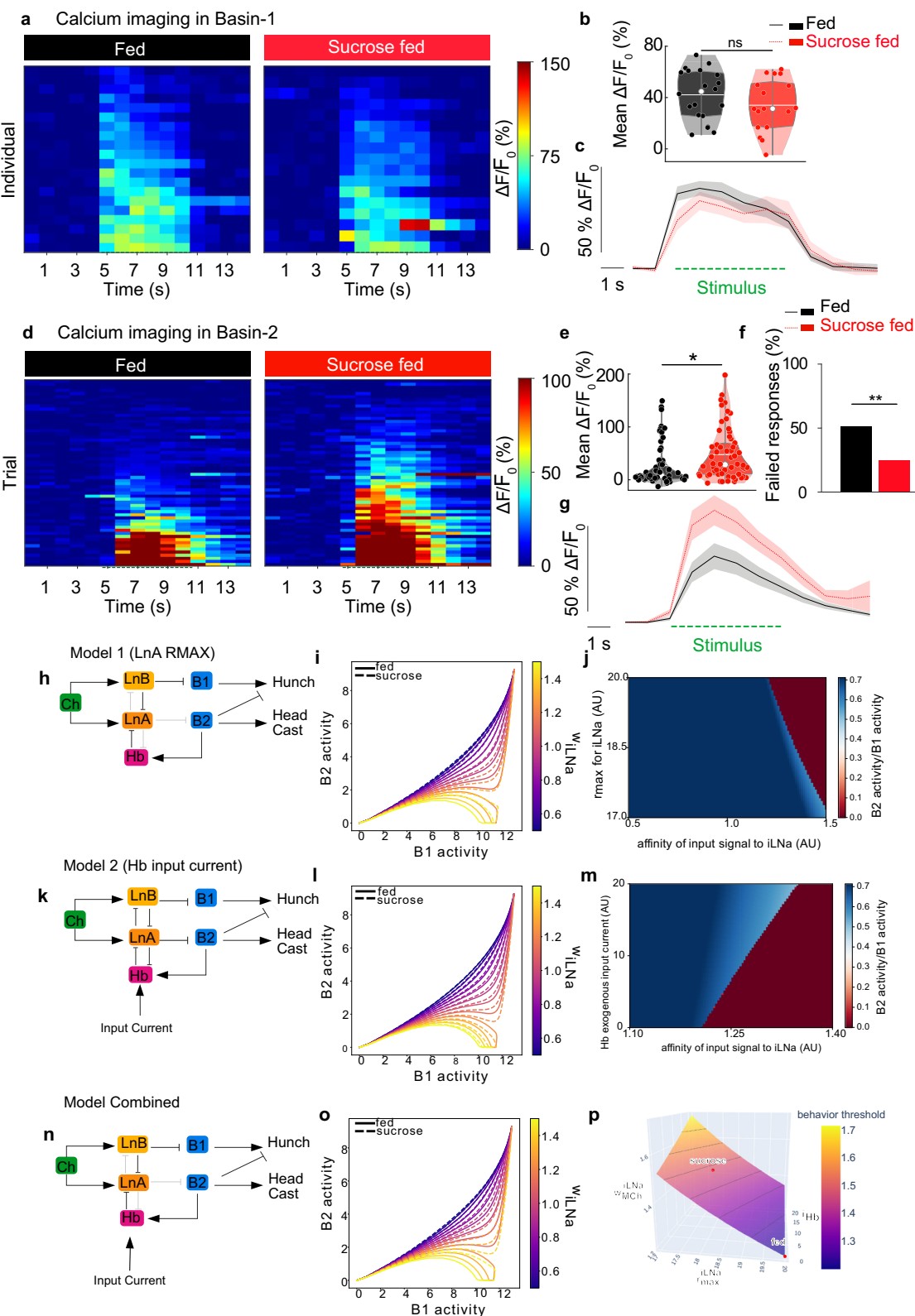

mechanical stimulation. In response to a moderately strong mechanical stimulus (5 V applied to the piezo at 1000 Hz), we observed a greater probability of the absence of a neuronal response (failed response) in normally fed larvae (Fig. 3f and Supplementary Fig. 3k). The previous study recorded depolarization responses simultaneously from Basin-1 and Basin-2 to a mechanical stimulus and showed that Basin-1 neurons always respond, whereas Basin-2 responses are probabilistic[32]. In this study, we computed the probability of Basin-2 responses to a mechanical stimulus and observed a significantly greater probability of responses in larvae fed sucrose only, which is consistent with higher Head Casting and lower Hunching probabilities in these larvae (Fig. 1n–q, Supplementary Fig. 1u–w). In starved larvae, similar trends were observed only at higher stimulus intensities (Supplementary Fig. 3l).

**Fig. 3 | The feeding status affects the output of the sensorimotor circuit.**
**a**–**c** Basin-1 (B1) calcium responses to mechanical stimulations in larvae fed standard food or sucrose (20B01-lexA;LexAop-GCaMP6s,UAS-CsChrimson-mCherry). **a** B1 individual calcium responses. **b** calcium response averaged during stimulation. **c** mean B1 calcium trace +/− SEM. **d**–**g** Basin-2 (B2) calcium responses to mechanical stimulations in fed and sucrose larvae (SS00739-UAS-GCaMP6s). **d** B2 individual calcium responses. **e** calcium response averaged during stimulation. **f** percentage of trials with failed responses (**g**) mean B2 calcium trace +/− SEM. **b**, **d**, White line: mean, white dot: median, colored dots: individual data points. **c**, **g** Green dashed line: stimulation. **h**–**p** Simple connectome-based rate model of the Hunch/Headcast selection circuit. **h**–**j** Model 1: sucrose state - decreased maximum LNa rate. **h** circuit schematic: weaker LNa influence on downstream targets (**i**) B1 and B2 Activity state-space trajectories as a function of MCh to LNa coupling. w_ILNA: output proportion from MCh feeding into LNa, 1-w_iLNa: proportion feeding into iLNb. Fed: rmax_-iLNa=20 a.u., Sucrose: rmax_iLNa=18 a.u. **j** phase diagram exploring multiple

rmax_iLNa values. The affinity of the input signal to iLNa is identical to w_iLNa. **k**–**m** Model 2: sucrose state: input current to Handle b (Hb) (**k**) circuit schematic: stronger Hb influence on downstream targets (**l**) B1 and B2 Activity state-space trajectories as a function of MCh to LNa coupling. Sucrose state: An Additional 10 a.u input current excites Hb. **m** phase diagram exploring multiple i_Hb values. The affinity of the input signal to iLNa is identical to w_iLNa. **n**–**p** Combined model: sucrose state - 10 a.u. input current to Hb and decreased Maximum LNa rate (18 a.u.). **n** circuit schematic, stronger Hb and weaker LNa influence on downstream targets (**o**) B1 and B2 activity state-space trajectories as a function of MCh to LNa coupling. **p** phase diagram. The threshold surface in the (rmax_iLNa, i_Hb, w_iLNa) space is shown. Gray lines: threshold levels (combinations of parameters achieving the same threshold). Red dots: fed and sucrose models simulated in o. (Statistics: **b**, **e** two-tailed T-test; **f** Chi-square (one-sided) test; ***$p < 0.001$, **$p < 0.01$, *$p < 0.05$). See also Supplementary Figs. 3 and 5 and Source Data 5 and 6.

## Reciprocally connected inhibitory interneurons in the decision circuit are differentially modulated by the feeding state

Based on the circuit model simulation for selecting between two air puff-induced actions (Hunch and Head Cast)[32], we predicted that the relative level of activation of the reciprocally connected feedforward inhibitory interneurons determines the outcome of the competition: Basin-1 only state (Hunch) or coactive state (Head Cast). In this study, we investigated whether changes in the feeding state affected the level of activation of two different classes of inhibitory neurons: feedforward and feedback inhibitory neurons.

We addressed this question by updating the rate model introduced in the previous study[32], where the Hunch-Head Cast circuit was characterized. In this model, connections between neuron classes were derived from EM data (Fig. 3h–p). Coupling coefficients are proportional to the number of synapses, whereas excitatory and inhibitory connections are treated separately. Each neuron category (mechano-ch, iLNa, iLNb, fbLN-Ha, and fLN-Hb) is reduced to a single population variable, and connections are simplified to reduce small differences in the synaptic count to unique values, thus reducing parameter choices (see previous study[32]). The rate vector of all units is evolved according to

$$\tau \frac{dr}{dt} = -V_0 - r + s + i + k_{ex}(r - r^{max})(A^{ex}r - A^{in}r) \qquad (1)$$

where $V_0$ sets the threshold for activation, s is the stimulus, i is an input from afferent populations, $r^{max}$ sets the maximum rate, and $A^{ex}_{ij}$ and $A^{in}_{ij}$ are the excitatory and inhibitory connection strengths from neuron j to neuron i, respectively. As previously described[32], the model predicts that the level of activation of the two classes of feedforward inhibitory neurons will determine the output state of the network at the level of the Basin-1 and Basin-2 neurons (Fig. 3h–p and Supplementary Fig. 4a).

We varied the parameter w_iLNa, which represents the balance between inputs from MCh to iLNa and inputs from MCh to iLNb. A higher w_iLNa corresponds to greater stimulation of iLNa and less stimulation of iLNb, and vice versa. We limited the source for the variation of the stimulus to this parameter. Based on the calcium imaging results (Fig. 3a–g and Supplementary Fig. 4g–p), we manipulated the activity state of the reciprocally connected feedforward and feedback inhibitory neurons in the circuit: LNa and Handle-b (Hb) neurons to explore whether the modulation of one of the neurons was sufficient to bias the state-dependent changes in neuronal activity and behaviors or the modulation of both neurons was required. In one version of the model, the sucrose state was modeled as the decreased maximum intensity of LNa neuron activity to explore the role of the feedforward inhibitory neurons in the feeding state modulation (Fig. 3h–j). Another version of the model was designed with the sucrose state represented by adding an input to the Handle-b neurons

to explore the contribution of feedback inhibitory neurons (Fig. 3k–m). Finally, we constructed a combined model in which both LNa and Handle-b neurons were modulated in the sucrose-fed state (Fig. 3n–p and Supplementary Fig. 4a–f). In all versions of the model, the Hunching decreased and Head Casting increased in the sucrose-fed state compared with the normal fed state (Fig. 3j, m, p), which is in good agreement with the experimental results.

Figure 3i, l, o shows samples of rate trajectories in the B1-B2 plane for each model. In accordance with the previous study[32], the rates of the B1 and B2 populations are attracted, depending on the value of w_iLNa, either to a coactivate state (low w_iLNa) or to a B1 high/B2 low state (high w_iLNa), not only for the fed models (solid lines) but also for the sucrose models (dotted lines). For each variant of the model, the dotted lines are above the solid lines of the same color, and the transition between the two states occurs at a higher w_iLNa value in the sucrose models than in the fed models.

We investigated the sensitivity of the threshold value to the magnitude of the parameter change defining the fed and sucrose states (Fig. 3j, m). The ratio of the steady-state activity of the B2 variable to the B1 variable is plotted for multiple values of the affinity of the input signal to iLNa, another name for w_iLNa, and of the parameter modeling the effect of food deprivation. The boundary between the blue and red areas corresponds to the threshold between the two end states as w_iLNa varies. In Fig. 3j, we show that as $r^{max}$ for iLNa decreases from 20 a.u. to 17 a.u., the threshold increases, as the boundary has a downward slope. Thus, the domain of inputs yielding a coactive response associated with Head Casts increases as the parameter modeling the effect of the diet change increases, in accordance with behavioral measurements. Similarly, in Fig. 3m, the threshold increases as the exogenous input current to Hb increases, albeit less dramatically. Once again, the region of stimuli that resulted in a Head Cast increased, which was consistent with the results of the behavioral experiments.

These model results suggest that modulating both feedforward and feedback inhibitory neurons of the circuit could result in the observed behavioral changes upon sucrose feeding. To test this experimentally, calcium transients in response to a mechanical stimulus were imaged in the two different types of inhibitory neurons to which we had genetic access (Fig. 4a–h and Supplementary Fig. 4g–p). An LNa neuron type, Griddle-2 (G2), which promotes Hunching[32], presents significantly lower responses to a mechanical stimulus in larvae fed sucrose than in normally fed larvae (Fig. 4a–d). A decrease in the response was observed at different intensities of mechanical stimulation (Supplementary Fig. 4m). Similarly, in starved larvae, the responses of Griddle-2 neurons also decreased across different intensities of stimulation (Supplementary Fig. 4g–i, n).

According to the model, a decrease in the activity of LNa neurons results in less Hunching and more Head Casting, which is consistent with the behavioral changes we observed after sucrose feeding. The

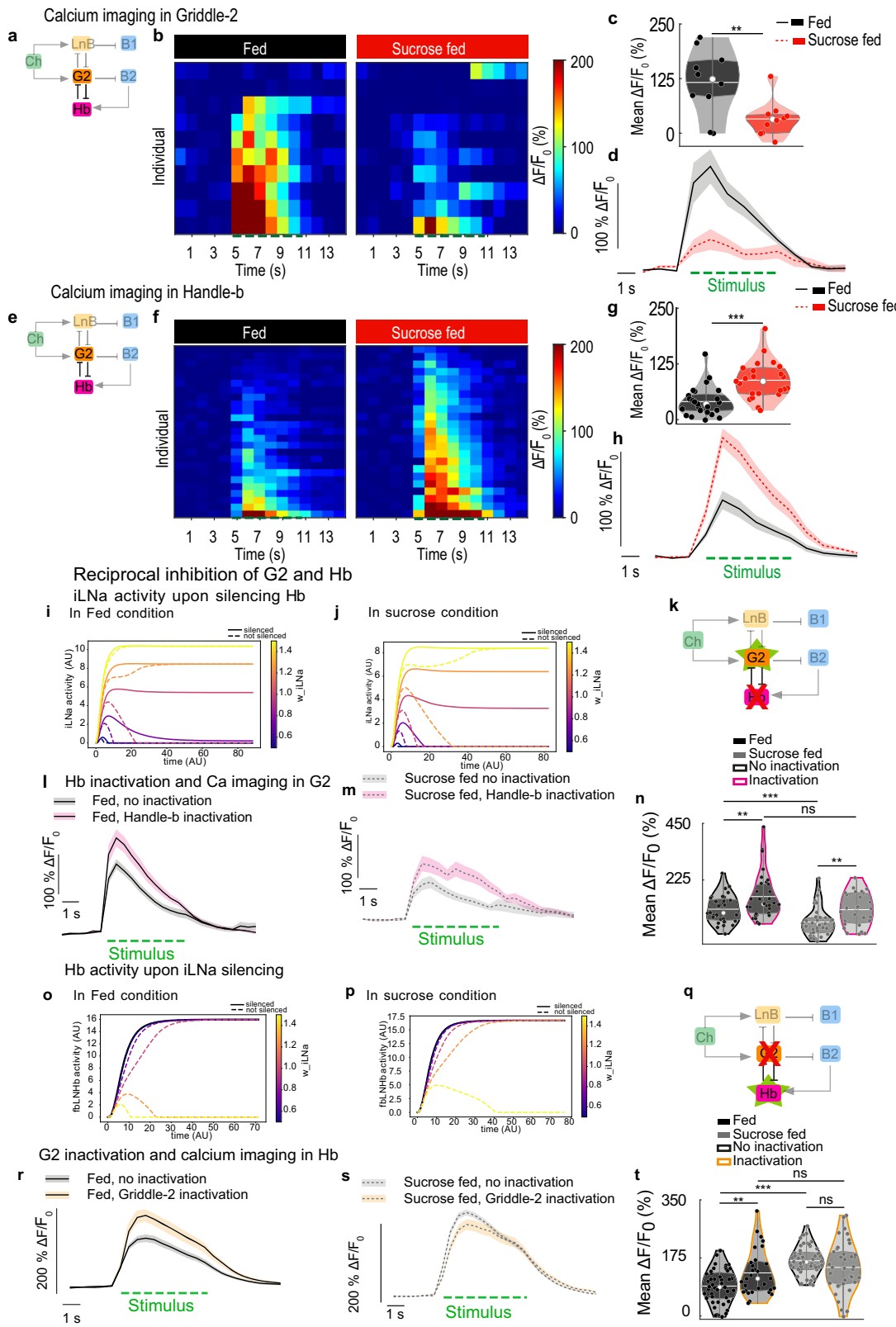

model also shows that another motif in the circuit, feedback disinhibition, promotes Head Casting by amplifying the activity of Basin-2. We recorded the responses of Handle-b neurons to the mechanical stimulus in larvae in the different feeding states and found that Handle-b neurons, which inhibit hunching[32], exhibit stronger responses to a mechanical stimulus in larvae fed sucrose and in starved larvae than in normally fed larvae (Fig. 4e–h and Supplementary Fig. 4j–l, o, p). This

result was observed for all the different intensities of stimulation we tested (Supplementary Fig. 4o, p). Therefore, the decreased response of Griddle-2 neurons (Fig. 4b–d) and increased response of Handle-b neurons (Fig. 4f–h) in larvae fed sucrose could bias the behavioral outcome, consistent with the behavioral changes observed in larvae upon sucrose feeding (less Hunching and more Head Casting) (Fig. 1n–q).

**Fig. 4 | Two reciprocally connected interneurons are oppositely modulated by the feeding state. a–d** Griddle-2 calcium responses to mechanical stimulations in larvae fed on standard food and sucrose (SS00918/UAS-GCaMP6s). **a** Imaging schematic. **b** Responses of different individuals, (**c**) mean calcium response averaged during the stimulus. **d** Mean Griddle-2calcium trace of over time +/− SEM. **e–h** Handle-b calcium responses to mechanical stimulations in larvae fed on standard food or on sucrose (SS00888/UAS-GCaMP6s). **e** Imaging schematic. **f** Responses of different individuals. **g** Mean calcium response averaged during the stimulus. **h** Mean Handle-b calcium trace over time +/− SEM. **i–t** Reciprocal inhibition between Handle-b and Griddle-2. **i, j** model simulation, trajectories of LNa activity upon Hb inactivation in fed (**i**, modeled as rmax_iLNa = 20 a.u.) and sucrose condition (**j**, modeled as rmax_iLNa = 18 a.u.) Silencing Hb increases activity in iLNa, transiently for low w_iLNa and at steady-state for high w_iLNa, more so in the sucrose than in the fed model. **k** Imaging schematic. **l, m** mean Griddle-2 calcium trace over time +/− SEM, with (SS0888-Gal4>UAS-TNT 55C05-LexA>LexAop-GCaMP6s) or without Handle-b inactivation (+/UAS-TNT 55C05-LexA>LexAop-GCaMP6s), in larvae fed on standard food (**l**) or on sucrose (**m**). **n** mean calcium response averaged during the stimulus. **o, p** model simulation, trajectories of Hb activity upon iLNa inactivation in fed (**o**, modeled as i_Hb = 0 a.u.) and sucrose condition (**p**, modeled as i_Hb = 10 a.u.). Silencing iLNa increases strongly the activity of Hb across w_iLNa values and for both models. **q** Imaging schematic. **r, s** mean Handle-b calcium trace over time +/− SEM, with (55C05-LexA > LexAop-TNT 22E09-Gal4>UAS-GCaMP6s) or without Handle-b inactivation (+/LexAop-TNT 22E09-Gal4 > UAS-GCaMP6s), in larvae fed larvae (**r**) or sucrose-fed larvae (**s**). **t** mean calcium response averaged during the stimulus. **c, g, n, t** White line represents the mean, white dot represents the median, colored dots represent individual data points. **d,h,l,m,r,s** The green dashed line corresponds to stimulation. (Statistics: **c, g, n, t** two-tailed T-test; ***$p < 0.001$, **$p < 0.01$, *$p < 0.05$). See also Supplementary Figs. 4 and 5. The source data and $p$-values are provided in Source Data 5 and 6.

The EM analysis revealed that Griddle-2 and Handle-b neurons are reciprocally interconnected (Fig. 4a)[32]. The circuit and the model predict that silencing Handle-b neurons results in an increase in Griddle-2 neuronal activity (Fig. 4i, j, and Supplementary Fig. 5a–f). Furthermore, the reciprocal connections between Griddle-2 and Handle-b neurons were probed functionally, and the model predictions were tested experimentally; calcium responses to a mechanical stimulus were monitored in Griddle-2 neurons while inactivating Handle-b neurons using tetanus toxin in both larvae that were fed standard food and those that were fed sucrose only. The responses were greater than those of the control larvae in both feeding states (Fig. 4k–n and Supplementary Fig. 5g), which is consistent with the inhibition of Griddle-2 neurons by Handle-b neurons. However, the increase in Griddle-2 responses upon Handle-b inactivation in larvae fed sucrose only did not reach the levels of Griddle-2 responses upon inactivation in larvae fed standard food (Fig. 4k–n and Supplementary Fig. 5g). This result could suggest that the increase in Handle-b activity after sucrose feeding is insufficient to reduce Griddle-2 responses in larvae fed sucrose and that Griddle-2 neurons could also be modulated by internal state signals. This finding is consistent with the predictions of Models 1 and 3, where Griddle-2 neurons are modulated extrinsically (Fig. 3h–j, n–p; Fig. 4i, j and Supplementary Fig. 5a–c).

The circuit and the model predict that silencing Griddle-2 neurons results in an increase in Handle-B activity (Fig. 4o, p). This prediction was tested by imaging Handle-b activity while inactivating Griddle-2 neurons (Fig. 4q–t). Handle-b responses to the mechanical stimulus were increased in normally fed larvae upon Griddle-2 inactivation (Fig. 4q–t, Supplementary Fig. 5h), which is consistent with the inhibition of Handle-b neurons by Griddle-2 neurons. In larvae fed sucrose, the activity of the Handle-b neuron was not significantly different upon Griddle-2 inactivation and was similar to the activity upon Griddle-2 inactivation in larvae fed standard food (Fig. 4q–t and Supplementary Fig. 5h). This lack of increase in Handle-b activity upon Griddle-2 inactivation in larvae fed sucrose could be due to the different state of the network in the larvae fed only sucrose: the activity of Griddle-2 neurons was already low in these larvae (Fig. 4b–d and Supplementary Fig. 4m), and their inactivation may not impact Handle-b neurons significantly.

**The feeding state-dependent increase in Basin-2 responses depends on changes in the activity of inhibitory interneurons**

Our previous work has shown that Basin-2 (B2) neurons receive input directly from the inhibitory neuron LNa and are disinhibited by the feedback inhibitory neuron Handle-b, which forms inhibitory connections with LNa and LNb neurons; the Handle-b neuron receives input primarily from Basin-2 neurons. The feedback disinhibition of Basin-2 neurons creates positive feedback that stabilizes Basin-2 ON state[32]. To determine whether the increase in Basin-2 responses in larvae fed on sucrose depends on feeding state-dependent changes in the activity of the inhibitory neurons, Handle-b neurons were optogenetically activated, and the activity of B2 neurons was recorded in response to a mechanical stimulus. We found that activating Handle-b in normally fed larvae increases the level of the Basin-2 response to a level similar to that observed in larvae fed sucrose (Fig. 5a–c and Supplementary Fig. 6a). We further silenced Handle-b neurons and monitored calcium responses in Basin-2 neurons in larvae with different feeding states (Fig. 5d–g and Supplementary Fig. 6b–d). Silencing of Handle-b neurons in larvae fed sucrose only decreased Basin-2 responses and erased the difference in Basin-2 response levels between larvae fed standard food and those fed sucrose only (Fig. 5d–g and Supplementary Fig. 6b–d). This result is consistent with the previous characterization of the Handle-b's disinhibition of Basin-2 neurons. Because Handle-b disinhibits Basin-2, silencing Handle-b therefore increases the inhibition of Basin-2 by LNa neurons and thus reduces the response of Basin-2 neurons to mechanical stimuli. In addition, in normally fed larvae, where the activity of LNa neurons is high, Basin-2 responses were almost completely abolished (Supplementary Fig. 6b–d).

Griddle-2 neurons were then inactivated, and the activity of Basin-2 neurons was imaged in larvae in the two feeding states. As expected, silencing Griddle-2 neurons abolished the difference in Basin-2 responses between the two states (Fig. 5h–k). This result was consistent with the effect of Griddle-2 neurons on Handle-b neuronal activity (Fig. 4o–t). Silencing Griddle-2 neurons in fed larvae resulted in a small (but not significant) increase in Basin-2 responses (Fig. 5i, j and Supplementary Fig. 6e, f). However, the mean response of Basin-2 neurons in larvae fed only sucrose unexpectedly and significantly decreased upon Griddle-2 silencing. We also computed Basin-2 response probabilities. While inactivating Griddle-2 did not affect Basin-2 response probabilities in normally fed larvae, it did affect Basin-2 response probabilities in larvae fed only sucrose; the non-response probabilities increased significantly upon Griddle-2 inactivation in sucrose-fed larvae (Fig. 5k and Supplementary Fig. 6h). The mean responses of trials with only Basin-2 ON responses reveal that the level of responses of Basin-2 was increased in normally fed larvae to the level found in larvae fed only sucrose in the control. The mean responses of trials with only Basin-2 ON responses also revealed that the level of the Basin-2 response decreased in larvae fed sucrose (Supplementary Fig. 6e, g) and was similar to the level of activity in normally fed control larvae.

Griddle-2 is thus required for state-dependent modulation of Basin-2, possibly by mediating the disinhibition of Basin-2 by Handle-b. Inactivating Griddle-2 in normally fed larvae disinhibits Basin-2. On the other hand, inactivating Griddle-2 in larvae fed on sucrose when the activity of Griddle-2 is already low (and Griddle-2 does not strongly inhibit Basin-2 neurons) may favor other inhibitory pathways and result in lower Basin-2 responses.

Handle-b activation and calcium imaging in Basin-2

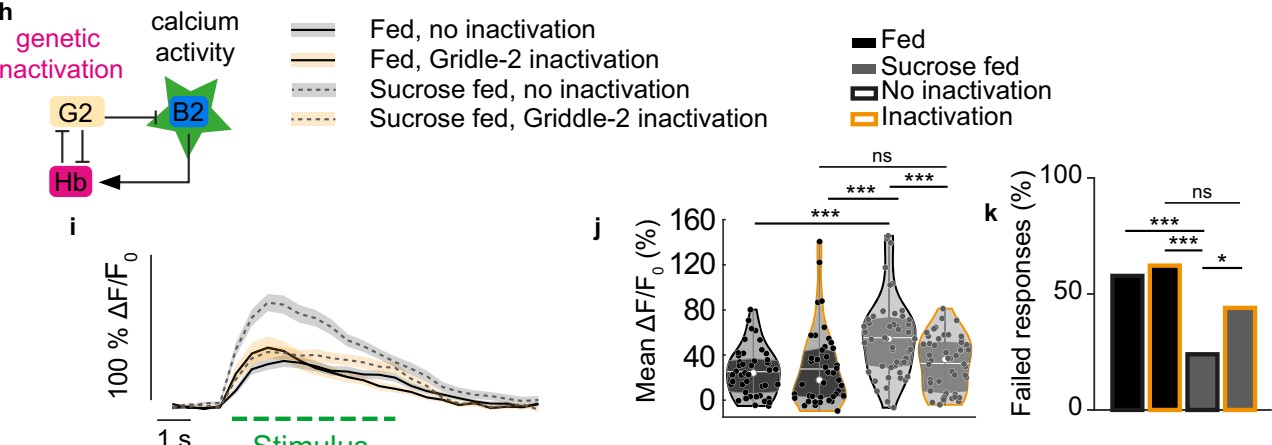

Handle-b inactivation and calcium imaging in Basin-2

Griddle-2 inactivation and calcium imaging in Basin-2

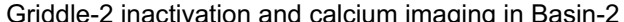

These results show that two types of reciprocally connected inhibitory neurons that promote competing actions inhibit each other during the response to a mechanosensory cue and are differentially modulated by changes in the feeding state (starvation and feeding on sucrose). The activity of Griddle-2 neurons, which promotes Hunching, is decreased, whereas the activity of Handle-b neurons, which inhibits Hunching and promotes Head Casting, is increased. This differential modulation of the inhibitory neurons, in turn, affects the state of the network in a feeding state-dependent manner and biases the behavioral responses toward

less Hunching and more Head Casting by increasing the activity of Basin-2 neurons when larvae are fed sucrose only.

**The feedback inhibitory neuron receives input from an NPF-releasing neuron that senses changes in the feeding state**
We examined the connectome of the upstream partners of Griddle-2 and Handle-b neurons to determine the sources of feeding state-dependent modulation. Previous work has shown that the different neurons in the circuit receive input from long-range projection neurons that, as suggested by their morphology and connectivity, could

**Fig. 5 | The modulation of inhibitory interneurons is required for the feeding state dependent changes in the circuit output. a** Imaging schematic. **b** mean calcium trace of Basin-2 over time +/− SEM from individuals fed on different diets, with or without optogenetic activation of Handle-b (22E09-Gal4>UAS-CsChrimson::tdTomato 38H09-LexA > LexAop-GCaMP6s) during the first second of mechanical stimulus (red box). The green dashed line corresponds to stimulus onset. **c** calcium response of Basin-2 averaged during the first second of the stimulus. The white line represents the mean, the white dot represents the median, and colored dots with white edges represent individual data points. Activating Handle-b in fed larvae phenocopies the effect of sucrose feeding on Basin-2 activity. **d** Imaging schematic. **e** mean calcium trace of Basin-2 over time +/− SEM from individuals fed on different diets, with (38H09-LexA > LexAop-GCaMP6s 22E09-Gal4 > UAS-TNT) or without (38H09-LexA > LexAop-GCaMP6s + / > UAS-TNT) Handle-b inactivation. The green dashed line corresponds to stimulus onset. **f** calcium response of Basin-2 averaged during the stimulus. The white line

represents the mean, the white dot represents the median, and colored dots with white edges represent individual data points. **g** percentage of trials that failed to elicit a calcium response in Basin-2. Inactivating Handle-b in larvae fed on sucrose only prevents the effect of sucrose feeding on Basin-2 activity. **h** Imaging schematic. **i** mean calcium trace of Basin-2 over time +/− SEM from individuals fed on different diets, with (38H09-LexA > LexAop-GCaMP6s 55C05-Gal4 > UAS-TNT) or without (38H09-LexA>LexAop-GCaMP6s + / > UAS-TNT) Griddle-2 inhibition. The green dashed line corresponds to stimulus onset. **j** calcium response of Basin-2 averaged during the stimulus. The white line represents the mean, the white dot represents the median, and colored dots with white edges represent individual data points. **k** percentage of trials that failed to elicit a calcium response in Basin-2. (Statistics: **c**, **f**, **j** one-way ANOVA with Tukey post-hoc test (two-sided); **g**, **k** Chi-square (one-sided) test; ***$p < 0.001$, **$p < 0.01$, *$p < 0.05$). See also Supplementary Fig. 6. The source data and $p$-values are provided in Source Data 5 and 6.

bias the output of the network based on contextual/internal state information[32]. Interestingly, by matching the light microscopy images with the reconstructed electron microscopy images, we found that one of these neurons is an NPF-expressing neuron that synapses on Handle-b neurons (Fig. 6a–c). NPF (neuropeptide F), a homolog of the mammalian neuropeptide Y[49], is a hunger signal, and its expression promotes feeding in both adult and larval *Drosophila*[50]. Therefore, we sought to investigate the role of NPF neurons in modulating the activity of inhibitory neurons in a state-dependent manner (Fig. 6 and Supplementary Fig. 7). Larvae have two pairs of NPF-expressing neurons. Both have cell bodies in the brain; the dorsolateral pair (DL-NPF) projects ipsilaterally within the brain lobes, and the dorsomedial pair (DM) sends descending projections via the suboesophageal zone (SEZ) through the ventral nerve cord (VNC) (Fig. 6a, b and Supplementary Fig. 7a–c).

To study the influence of NPF neurons on Handle-b activity, the activity of the DM-NPF descending neuron was monitored in larvae in the different feeding states. Therefore, we drove GCaMP6s expression in the NPF descending neurons with NPF-GAL4 and measured its fluorescence intensity in the VNC projections of larvae fed either standard food (fed), 20% sucrose (sucrose-fed), or completely starved (administered only water for 90 minutes). We showed that starvation or feeding with sucrose both increased the activity of the NPF descending neurons (Fig. 6d–f). The application of glucose to NPF neurons decreased their activity in the presence of tetrodotoxin (TTX), suggesting that they may sense glucose autonomously (Supplementary Fig. 8a).

NPF was shown to be involved in promoting feeding[51]. To investigate whether the increase in activity of NPF descending neurons upon 90-minute sucrose feeding or starvation had an influence on NPF release and behavior, NPF was knocked down selectively in the NPF descending neuron using a split-GAL4 driver that labels the descending pair of NPF neurons (Fig. 6a) and larval locomotion was monitored in larvae that were fed standard food, larvae that were fed sucrose and starved larvae (Fig. 6g, h and Supplementary Fig. 8b–g). The downregulation of NPF in DM-NPF neurons descending into the VNC impaired the increase in exploration observed in larvae fed sucrose and starved larvae (Fig. 6g, h and Supplementary Fig. 8b–g).

To understand the influence of NPF neurons on Handle-b neuronal activity, the interactions between the two neurons were tested functionally by hyperpolarizing the NPF neurons with the inwardly rectifying potassium channel Kir2.1 using an NPF-LexA driver line that labels both pairs of NPF neurons and monitoring the calcium responses of Handle-b neurons to mechanical stimulation. Upon NPF neuron silencing, we observed a decrease in Handle-b responses to mechanical stimulation (Fig. 6i–k and Supplementary Fig. 8h–m). The NPF descending neuron could thus facilitate the responses of Handle-b neurons. Upon changes in the feeding state, the activity of NPF neurons increases upon starvation and sucrose feeding, which could enhance

Handle-b neuronal responses to a mechanical stimulus, resulting in greater responses of these neurons (Fig. 6d−f, i−k) that in turn would bias behavioral choice toward less Hunching and more Head Casting.

In the connectome, we observed large dense core vesicles in NPF neurons in proximity to Handle-b neurons (Fig. 6c). Moreover, we showed that Handle-b neurons express NPFR1 using the T2A-LexA method[52], where we drove the expression of one reporter in all NPFR1 neurons using NPFR1 T2A-Lexa and another reporter with a Handle-b-specific split-GAL4 driver. The intersection of the two expression patterns revealed that Handle-b neurons express NPFR1 (Fig. 6l–p). The model and functional connectivity experiments predict that Handle-b and Griddle-2 neurons could be independently modulated (Fig. 4i–t and Supplementary Fig. 5a–f). We thus investigated whether Griddle-2 neurons could also be influenced by NPF. We found that Griddle-2 neurons do not receive synaptic input from the NPF descending neurons. Since neuropeptides can act outside of synaptic sites and we found dense core vesicles characteristic of neuropeptide release along the NPF axon in proximity to Griddle-2 neurons, we also examined NPFR1 expression in Griddle-2 neurons, as well as in Basin-1 and Basin-2 neurons, using a similar approach as that used for Handle-b neurons and did not observe any NPFR1-positive cells among these neurons (Supplementary Fig. 7f–h).

We then downregulated the expression of NPFR1 in Handle-b neurons and imaged their calcium responses to mechanical stimulation in larvae fed standard food, larvae fed only sucrose, and starved larvae (Fig. 6q−s and Supplementary Fig. 8n, o). The downregulation of NPFR1 in Handle-b neurons abolished its feeding-state-dependent modulation as a significant increase in Handle-b responses was not observed upon sucrose feeding or starvation when NPFR1 was downregulated (Fig. 6q−s and Supplementary Fig. 8n, o). This outcome was reflected in the behavioral responses to an air puff, where upon NPFR1 knockdown in Handle-b neurons, no differences in behavioral responses could be observed between the three different states, i.e., fed, sucrose only and starved, further suggesting that NPF signaling is required for the state-dependent modulation of these neurons (Supplementary Fig. 9).

Taken together, these results show that NPF neurons sense changes in the feeding state and that their activity increases upon starvation or sucrose-only feeding. The changes in NPF activity influence Handle-b neuronal activity, which increases upon starvation or sucrose-only feeding due to the facilitating effect of NPF. This increase in activity would then bias the sensorimotor decisions toward less Hunching and more Head Casting.

### sNPFR in interconnected inhibitory interneurons mediates the state-dependent modulation of larval behavioral responses to a mechanical stimulus

The model and calcium imaging results suggest that Griddle-2 and Handle-b neurons are independently modulated by the feeding state

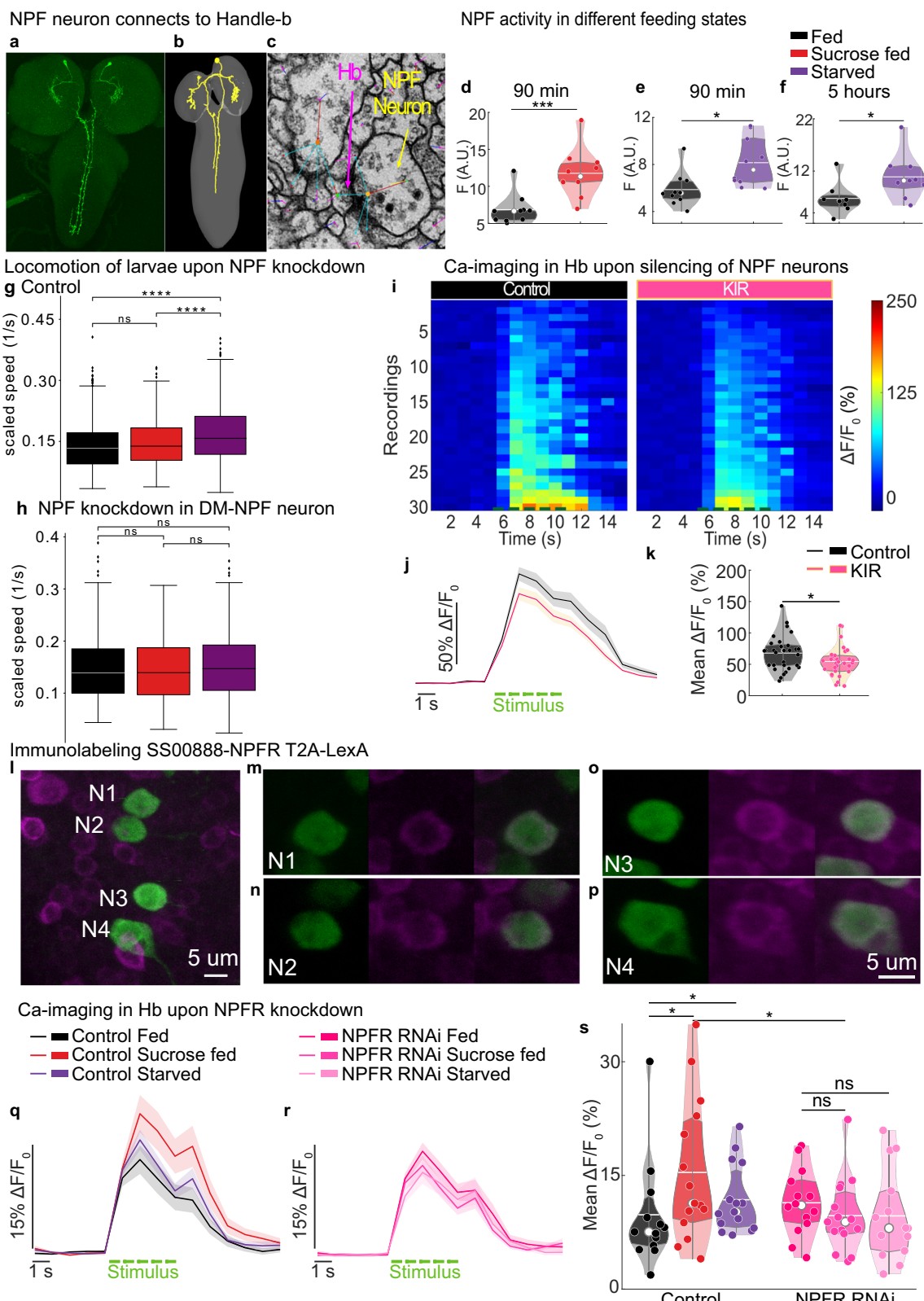

**NPF neuron connects to Handle-b**

a    b    c

**NPF activity in different feeding states**

d  90 min    e  90 min    f  5 hours

- Fed
- Sucrose fed
- Starved

**Locomotion of larvae upon NPF knockdown**

g  Control

**Ca-imaging in Hb upon silencing of NPF neurons**

i

h  NPF knockdown in DM-NPF neuron

j    k

**Immunolabeling SS00888-NPFR T2A-LexA**

l    m    o    n    p

**Ca-imaging in Hb upon NPFR knockdown**

- Control Fed
- Control Sucrose fed
- Control Starved
- NPFR RNAi Fed
- NPFR RNAi Sucrose fed
- NPFR RNAi Starved

q    r    s

(Fig. 4). Since Griddle-2 neurons do not express the NPFR1 receptor, other neuropeptides could influence Griddle-2 and the circuit. We therefore investigated the effect of another neuropeptide on the circuit: sNPF (the short neuropeptide F). sNPF is a second homolog of the mammalian NPY, whose receptor is widely distributed in the larval nervous system, including the VNC[53], and was shown to be involved in regulating hunger-driven behaviors[50] and facilitating mechano-

nociceptive responses[40], among other functions. Therefore, we sought to examine whether the interneurons in the circuits express the receptor for sNPF (Fig. 7 and Supplementary Fig. 7). By genetically coexpressing a red fluorescent reporter (jRGECO1a) under the control of LexA drivers that label either the Handle-b, Griddle-2, Basin-1 or Basin-2 neurons we identified in the literature and existing Gal4 databases (see Methods for details) and a green fluorescent reporter

**Fig. 6 | NPF-releasing neurons convey internal state information to the feedback inhibitory neuron. a, b** Comparison of light microscopy images (GMR_SS01635 > UAS-GFP) with electron microscopy (EM) reconstruction images of DM-NPF neurons. **c** EM image shows neuropeptide-containing dense core vesicles in the DM-NPF neuron near one of its synapses with Handle-b. Red arrows: presynaptic neurons, DM-NPF annotated with yellow arrow, blue arrows: postsynaptic sites, magenta arrow: Handle-b **(d–f)** Baseline calcium fluorescence in DM-NPF ventral nerve cord projections upon 90 min sucrose feeding **(d)**, starvation for 90 min **(e)** or 5 hours **(f)**. **g, h** Locomotion analysis. Box plots display the median, interquartile range (IQR), and whiskers up to 1.5 × IQR. Outliers are plotted individually. **g** control larvae. **h** NPF knockdown in DM-NPF neurons. **i–k** Handle-b calcium responses with (UAS-GCaMP6s; 22E09-Gal4, NPF-LexA > LexAop-KIR) or without (UAS-GCaMP6s; 22E09-Gal4, NPF-LexA;+) DM-NPF silencing in fed larvae **(i)** Handle-B calcium responses, single trial of mechanosensory stimulation. **j** mean Handle-B calcium trace over time +/- SEM. **k** mean calcium response averaged during the stimulus. **l–p** Immunohistochemical labeling for NPFR in Handle-b (all images, scale bar = 5 μm). GMR_SS00888 split-Gal4 drives UAS-GCaMP6s in Handle-b, and LexAop-jRGECO1a is expressed under the control of the NPFR promoter using a T2A-LexA construct (magenta). GFP and dsRed antibodies increase detection sensitivity. **l** 4 neurons (N1-N4), 23 μm stack. The **(m–p)** whole thickness (5 to 7 μm) of each neuron (N1-N4) is shown. **q–s** Handle-B calcium responses upon NPFR knockdown (GMR_SS00888 > UAS-NPFR-RNAi; UAS-GCaMP6s) compared to control (GMR_SS00888 > UAS-GCaMP6s). **q** mean Handle-b calcium trace over time +/- SEM in control larvae fed, fed on sucrose or starved. **r** mean Handle-b calcium trace over time +/- SEM upon NPFR knockdown in larvae fed, fed on sucrose or starved. **s** mean calcium response averaged during the stimulus. **f, k, s** White line: mean, white dot: the median, colored dots: individual data points. **j, q, r** green dashed line: stimulus. (Statistics: **d–f, k** two-tailed T-test; **g, h** two-sided Mann-Whitney test with Bonferroni correction; **s** one-sided Mann-Whitney test; ****$p < 0.0001$, ***$p < 0.001$, **$p < 0.01$, *$p < 0.05$). See also Supplementary Figs. 7 and 8. The source data and $p$-values are provided in Source Data 1, 5 and 6.

(GCaMP6s) under the control of the sNPFR1 promoter with a T2A-Gal4 construct, we showed that both Handle-b (Fig. 7a–d) and Griddle-2 (Fig. 7l–o) neurons express sNPFR1, whereas the two Basin neurons do not (Supplementary Fig. 7i, j).

We determined whether sNPF signaling was involved in the response of these two types of neurons to a mechanical stimulus by monitoring their responses upon sNPFR1 downregulation. Calcium imaging recordings revealed that genetically downregulating sNPFR1 expression in Handle-b neurons increased their response to mechanical stimulation (Fig. 7e–g), suggesting that sNPFR1 has an inhibitory effect on Handle-b neurons. On the other hand, the responses of Griddle-2 neurons were decreased upon sNPFR1 downregulation (Fig. 7p–r), suggesting that sNPF facilitates Griddle-2 neuronal responses to a mechanical stimulus. Since Handle-b neurons inhibit Hunching and promote Head Casting, while Griddle-2 neurons promote Hunching and inhibit Head Casting, downregulating the sNPFR1 in these neurons would inhibit Hunching and promote Head Casting. Indeed, the behavioral experiments showed that sNPFR1 knockdown in either of these two types of neurons led to less Hunching, whereas sNPFR1 knockdown in Griddle-2 neurons resulted in significantly more transitions to Head Casting in response to an air puff (Fig. 7h–k, s–v).

sNPFR1 knockdown in Handle-b and Griddle-2 neurons leads to a modulation of their activity similar to that caused by feeding on only sucrose or starvation (lower Griddle-2 activity and higher Handle-b activity) and yields the same behavioral outcome. These findings suggest that sNPF signaling, which targets these neurons, may be downregulated in larvae fed sucrose and starved larvae and at least partially responsible for the modulation of the behavioral choices of the larvae in response to mechanical stimulation. In line with this hypothesis, the downregulation of sNPFR1 in Handle-b or Griddle-2 neurons did not significantly impact their calcium or behavioral responses to mechanical stimulation in larvae fed sucrose or starved (Supplementary Figs. 10 and 11).

Taken together, these results show that Handle-b and Griddle-2, reciprocally interconnected inhibitory neurons that drive opposing actions, are differentially modulated by the sNPF signaling pathway to bias the response to an air puff toward less Hunching and more Head Casting.

## Discussion

Although many studies have shown that various behaviors are affected by an animal's physiological state, especially hunger, this work goes further by describing the neural circuit mechanisms by which transient changes in the physiological status impact behaviors that are not directly related to feeding itself. We took advantage of the well-characterized behavioral response of *Drosophila* larvae to a mechanical stimulus for which we previously dissected the circuit mechanisms underlying the selection between two main types of responses: Hunch and Head Cast, to determine whether and how changes in the feeding state affect behaviors unrelated to feeding[32]. The current study reveals the neural circuit mechanisms of the modulation of these sensorimotor decisions by feeding conditions. Slight changes in feeding conditions affect an animal's motivational state and bias responses to an air puff toward less Hunching and more Head Casting. Reciprocally interconnected inhibitory neurons that drive competing actions are differentially modulated by changes in the feeding state: the activity of the neuron that promotes a Hunch is decreased, and the activity of the neuron that inhibits a Hunch is increased. This modulation at the level of inhibitory neurons involving the NPF and sNPF signaling systems biases the output of the network toward promoting Head Casting and inhibiting Hunching, a modulation that is consistent with the state-dependent behavioral changes we observed.

Hungry animals that behave differently in nonfeeding-related contexts have been reported in various studies and across the animal kingdom[7,54]. However, the underlying logic and neural mechanisms are not well understood. These changes could be linked to overall changes in the behavioral strategies of hungry individuals[7] to ensure survival. Risk-taking has been shown to be increased by hunger[54–56], even in social-related decisions[7,57], further raising questions about how food deprivation signals are integrated into the neural computations underlying nonfeeding-related behaviors. Animals need to be able to make any decisions flexibly depending on the need that is most critical at a given moment. Our study revealed that a circuit underlying sensorimotor decisions in response to a mechanical stimulus located at the early stages of sensory processing is influenced by changes in diet. This finding supports the idea that state-dependent flexibility of behavior could be achieved by the physiological state acting on circuits throughout the nervous system and thus reorganizing its activity in a distributed manner. In various ecological contexts, survival often involves a trade-off between avoiding danger and pursuing food- and water-seeking behaviors[54]. For example, increased exploration increases the likelihood of finding food but also of encountering a dangerous situation[54]. If animals' need for food or water increases, either due to food deprivation or thirst, they might be more likely to explore intensively despite an increasing risk of threat. Similarly, food-deprived and thirsty animals might take more risks and ignore aversive cues to increase their chances of accessing food or water sources. Hungry animals use different strategies when escaping predators than do satiated animals[17,23].

Monitoring locomotion in completely starved larvae or larvae fed sucrose revealed similar strategies of increased exploration in these two groups of larvae compared with normally fed larvae. These findings suggest that, despite the differences in the type of feeding state and glucose levels, the effects of nutrient deprivation on behavior were largely similar. This result could reflect an increased motivational drive to search for the nutrients the larvae are lacking, regardless of the type

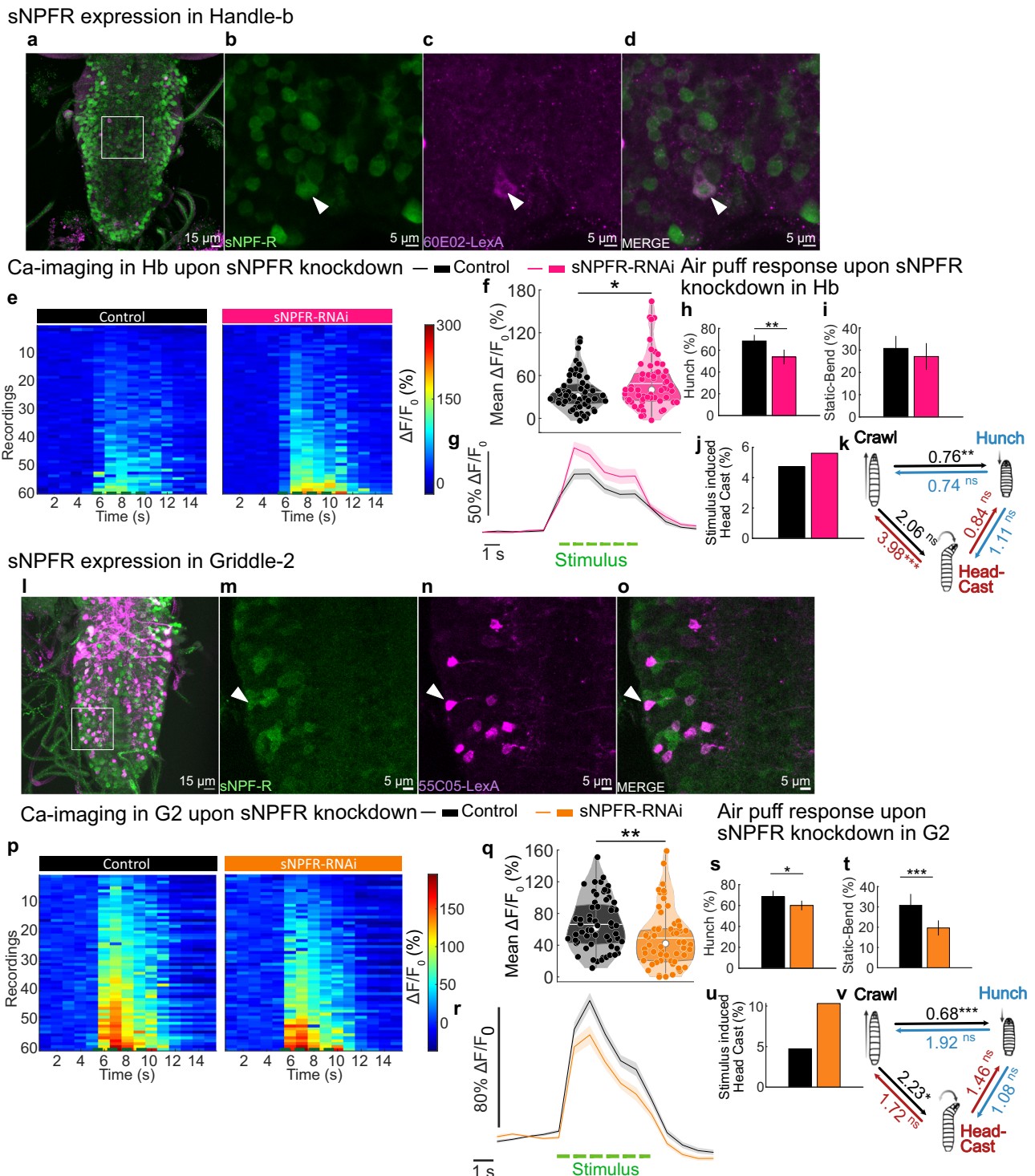

of nutrients missing – it could be water in the case of sucrose-fed larvae and any food in the case of starved larvae. This increased motivation to find the missing nutrients would then translate into similar changes in behavioral strategies.

Moreover, we surprisingly observed that, compared with complete starvation, sucrose feeding seemed to have a stronger effect on neuronal activity and behavioral changes in response to mechanical stimuli. Specifically, the effects of starvation on Griddle-2 and Basin-2 neurons were weaker, resulting in, for example, mildly increased activity of Basin-2 neurons only at higher stimulus intensities compared with increased activity across all intensities for Basin-2 neurons. This difference in Basin-2 responses between larvae fed only sucrose

and starved larvae could explain why Hunching was not consistently significantly reduced in starved larvae, whereas it was only significantly reduced in sucrose-fed larvae.

The stronger effects of sucrose-only feeding on neuronal activity and behavior could be due to dehydration being a more critical state than hunger (after 90 min of treatment). After prolonged treatment with 20% sucrose (> 5 h), larvae start dying, whereas starved larvae continue to develop. Alternatively, the lack of water and protein and increased glycemia in sucrose-fed larvae could all contribute to stronger immediate effects on neuronal activity and behavior.

The similar effects of the two feeding states on neuronal activity and behavior point to shared pathways and mechanisms that convey

**Fig. 7 | sNPF signaling inhibits Handle-b and facilitates Griddle-2 mechanosensory responses.** **a**–**d** Immunohistochemistry labeling for sNPFR in Handle-b. 60E02-LexA drives LexAop-jRGECO1a in Handle-b (magenta), and UAS-GCaMP6s is expressed under the control of the sNPFR promoter using a T2A-Gal4 construct (green). Antibodies against GFP and dsRed were used to increase detection sensitivity. **e**–**g** Calcium responses in Handle-b upon sNPFR knockdown (GMR_SS00888 > UAS-GCaMP6s; UAS-sNPFR-RNAi) compared to a control (GMR_SS00888 > UAS-GCaMP6s) in Handle-b. **e** calcium responses of Handle-b from different individuals and trials. One line is a trial. **f** mean calcium response averaged during the stimulus. **g** mean calcium trace of Handle-b over time +/− SEM. **h**–**k** Behavior in response to air-puff upon sNPFR knockdown in Handle-b (SS00888 > sNPFR-RNAi) compared to the control. **l**–**o** Griddle-2 immunostaining for sNPFR expression. 55C05-LexA line drives LexAop-jRGECO1a in Griddle-2(magenta), and UAS-GCaMP6s is expressed under the control of the sNPFR promoter using a T2A-Gal4 construct (green). Antibodies against GFP and dsRed were used to increase detection sensitivity. **p**–**r** Calcium responses in Griddle-2 upon sNPFR knockdown (SS_TJ001 > UAS-GCaMP6s; UAS-sNPFR-RNAi) compared to the control (SS_TJ001 > UAS-GCaMP6s). **p** calcium responses of Griddle-2 from different individuals and trials. One line is a trial. **q** mean calcium response averaged during the stimulus. **r** mean calcium trace of Griddle-2 over time +/− SEM. **s**–**v** Behavior in response to air-puff upon sNPFR knockdown in Griddle-2 (SS_TJ001 > sNPFR-RNAi) compared to the control. **f**, **q** White line represents the mean, white dot represents the median, colored dots represent individual data points. **g**, **r** The green dashed line corresponds to the stimulus. **h**–**j**, **s**–**u** Hunch and Static-bend are presented as population probability and 95 % confidence interval, Head Cast as post-stimulus probability corrected by baseline probability. **k**, **v** Behavioral transitions over the first ten seconds of stimulation (Maximum likelihood test (one-sided, chi-square approximation)). (Statistics: **f**, **q** two-tailed T-test; **h**, **i**, **s**, **t** Chi-square (one-sided) test; **j**, **u** Numerical simulation test; **k**, **v** Maximum likelihood test (one-sided, chi-square approximation); ***$p < 0.001$, **$p < 0.01$, *$p < 0.05$). See also Supplementary Figs. 7, 9. The source data and p-values are provided in Source Data 2, 3, 5 and 6.

information about the changes in the feeding state to the circuitry underlying locomotion and sensorimotor decisions. Thus, different neurons and circuits that sense different types of states could converge onto the same descending neurons that modulate the activity in the VNC circuits that control different behaviors.

While various studies have shown that hunger affects behavior by altering the responses of sensory neurons[9,11,58,59], others have implicated central mechanisms in feeding state-dependent behavioral flexibility[13,31,60]. We found that the activity of chordotonal mechanosensory neurons that sense an air puff was not significantly altered upon sucrose feeding or starvation, in contrast to the activity of downstream neurons. The changes at the level of sensory pathways tune animals' perception to increase their likelihood of finding food and feeding by increasing their responsiveness to appetitive stimuli and decreasing their responsiveness to aversive food-related stimuli. The implication of central mechanisms, on the other hand, may suggest that hunger acts as a global regulator of behavior, i.e., hunger may change brain activity in such a way that goals and behavioral strategies can be reevaluated to increase the animals' chances of survival[7]. Our experiment monitored the calcium responses of all the chordotonal sensory neurons together. The different inhibitory and projection neurons in the circuit receive inputs from different chordotonal subtypes. This finding could also explain the differential modulation of the different neuronal subtypes in the circuit if some subtypes of chordotonal neurons are modulated by changes in the internal state, while others are not. However, the optogenetic activation of all the chordotonal neurons under the different feeding conditions still resulted in decreased Hunching and increased Head Casting upon starvation or the consumption of a sucrose diet, strongly suggesting that downstream neurons are involved. Even if chordotonals are involved in altering behavior in a state-dependent manner, their modulation is not required to alter the behavior in response to the mechanical stimulus due to the contribution of the downstream circuitry.

Calcium imaging combined with neuronal manipulations revealed that in the circuit for selecting between the Hunch and the Head Cast, inhibitory neurons (and not projection neurons) were the target of modulation by changes in the feeding conditions. Inhibitory neurons were shown to be the target of contextual modulation in other systems[61–63]. Previous work has identified the reciprocal inhibition of inhibition as a motif underlying the competition between a startle-type action and an exploratory action[32,42]. This motif was shown to underlie similar computations in different species and brain areas[32–35,64–66] and was proposed to provide flexibility to the selection process[32,33,35]. The current study revealed that one of the reciprocally connected inhibitory neurons within this motif, LNa type Griddle-2 neurons, is modulated by changes in the feeding conditions and that this modulation contributes to biasing sensorimotor decisions. These results confirm theoretical predictions that such a motif confers the sensorimotor circuit the capacity to be tuned to other types of information, in this case, internal state information (starvation and thirst). Moreover, these findings support predictions that shaping the output of the network through disinhibition by reciprocally connected inhibited neurons allows for flexible, competitive selection[32,33,35].

This work revealed that another type of inhibitory neuron, Handle-b, which is a feedback inhibitory neuron that provides positive feedback to the Hunch inhibiting Basin-2 neurons through feedback disinhibition, is also modulated by changes in internal physiology. Handle-b neurons are also reciprocally connected to the LNa neurons that participate in the reciprocal inhibition of inhibition motifs. This connectivity pattern suggests that, in addition to competition within the reciprocal inhibition of inhibition motifs, the competition between the two layers of the circuits is also modulated by changes in internal physiology. Similar to recurrent excitation, the feedback disinhibition motif provides positive feedback that stabilizes the selected output[32]. Using inhibitory connections rather than excitatory connections may have the advantage of allowing decisions to be influenced by contextual and state information. In addition, in this circuit architecture, the feedback inhibitory neuron Handle-b contacts both reciprocally connected inhibitory neurons in the circuit and can thus shape the circuit activity at the level of the site of competition. It is thus well suited to integrate mechanosensory information and information about an animal's state.

Thus, two different layers of the network are modulated by changes in the feeding state, reciprocally connected feedforward inhibitory neurons and feedback inhibitory neurons that stabilize the state of the network, resulting in the coactivation of both Basin-1 and Basin-2 neurons.

The behavioral responses of the larvae to a mechanical cue depend on the state of the circuit at the level of reciprocally interconnected inhibitory neurons, which will shape the activity of the Basin projection neurons to either give rise to a state where Basin-1 only is active or a state where Basin-1 and -2 are both active[32]. We showed that the feeding state-dependent modulation of Basin-2 neurons depends on the modulation of the activity of Handle-b and Griddle-2 neurons. Accordingly, genetic colabeling of Basin-1 or Basin-2 neurons with either sNPFR1 or NPFR1 using T2A GAL4/LexA technology revealed no expression of these neuropeptide receptors in the two Basin neurons. This result is consistent with the finding that the state-dependent changes in the air puff-induced sensorimotor decisions are caused by the modulation that acts on the reciprocally interconnected inhibitory neurons that integrate mechanosensory information with current internal state needs.

Previous work revealed that the inhibitory neurons in the mechanosensory circuit are in contact with long-range projection neurons[32]. These long-range projection neurons may carry contextual

or state information to the mechanosensory circuit. We found that among these long-range projection neurons is a pair of NPF-releasing neurons that contact Handle-b neurons. NPF is a homolog of the mammalian neuropeptide Y, which is a hunger signal. NPF neurons have indeed been shown to be involved in hunger-dependent behaviors in both adult flies and larvae[50,51]. We found that the activity of the NPF descending pair of neurons changes as a function of the feeding state. These neurons could thus convey information about the satiation state to the mechanosensory circuit and bias sensorimotor decisions to mechanosensory cues by modulating the activity of Handle-b neurons. Additionally, the release of NPF at the proximity of the circuit by the NPF neurons (as suggested by the existence of dense core vesicles) could modulate Handle-b neurons, which express the NPFR1 (while other neurons in the circuit do not). The fact that the feedback inhibitory neurons are directly influenced by the NPF neurons supports the idea that it could gate circuit activity in a state-dependent manner. Our results indicate that both the feeding state-induced changes in the locomotor strategy (motivational locomotor state and exploration persistence) and the acute response to transient environmental stimuli are dependent on NPF signaling. The descending NPF neurons could convey information about the internal state to diverse neuronal populations in the VNC to regulate exploratory locomotion and stimulus-dependent motor responses, thus adjusting behavioral interactions with the environment according to the animal's current needs. NPF could serve as an internal state signal that couples various sensorimotor behaviors to the motivational/exploratory state of the animal and its physiological needs, thus regulating behavior across different timescales.

## Methods

### *Drosophila* rearing and handling

Flies (*Drosophila Melanogaster*) were raised on a standard food medium (ethanol 2%, methylhydroxybenzoate 0.4%, yeast 8%, cornmeal 8%, and agar 1%) at 18 °C. Third instar larvae were collected as follows: male and female flies from the appropriate genotypes were placed together for mating, then transferred at 25 °C for 12–16 h on a petri dish containing a fresh food medium for egg laying. The petri dish was then placed at 25 °C for 72 h. Foraging third instar larvae were collected from the food medium by using a denser solution of 20% sucrose, scooped with a paint brush into a sieve, and gently and quickly washed with water. Larvae used for optogenetic experiments were raised at 25 °C in complete darkness, on standard food supplemented with all-trans retinal at 0.25 mM (R240000, Toronto Research Chemicals). The full list of genotypes used in the study can be found in the Supplementary Table 1 Resource table.

### Dietary treatments

For dietary treatments, larvae were placed in 60×15 mm circular petri dishes that contained a 45 mm circular Whatman paper. Larvae were subjected to different diets: standard food without agar for fed larvae (as described in the *Drosophila* rearing section), 20% sucrose solution for sucrose fed larvae, standard food without agar prepared in 20% sucrose solution for food+sucrose fed larvae, 23.2% sucralose solution for sucralose fed larvae, and water for starved larvae. The Whatman paper was soaked with 0.6 mL MilliQ water (starved), sucrose solution (fed on sucrose), or soaked with 0.6 mL water, and 1 mL of standard food medium was added on top (fed). Larvae were collected after the appropriate amount of time (90 or 300 min) and rinsed in water before behavioral, imaging, or biochemistry experiments. For the behavioral experiments with rehydration, larvae were collected after 90 min, rinsed in water, and placed in a petri dish with a Whatman paper soaked with 0.6 mL water for 15 min. Likewise, for the refeeding experiments, starved larvae were collected after 300 min, rinsed, and placed for 15 min in a petri dish with

standard food medium and water as in the fed condition. After the treatment, larvae were once more collected and rinsed before the experiment.

### Behavioral tracking

We used an apparatus previously described[47,48]. Briefly, the apparatus comprises a video camera (Basler ace acA2040-90 μm) for monitoring larvae, a ring light illuminator (Cree C503B-RCS-CW0Z0AA1 at 624 nm in the red), a computer, and a hardware module for controlling air-puff, controlled through multi worm tracker (MWT) software (http://sourceforge.net/projects/mwt)[48,67]. The arena consisted of a 25625 cm2 square of 3% Bacto agar gel (CONDALAB 1804-5) with charcoal (Herboristerie Moderne, 66000 Perpignan) in a plastic dish, and was changed for each experiment. For optogenetic experiments, plates without charcoal were used, and larvae were tracked thanks to IR light. Collected third instar larvae were washed with water, moderately dried, and spread on the agar starting from the center of the arena. We tested ~ 30–100 larvae at once during each experiment. The temperature of the behavioral room was kept at 25 °C.

### Locomotion analysis

To assess larval locomotion in the absence of stimulation, larvae were placed on top of the agar in the arena inside the tracker, and either tracked for 5 min continuously (intact attP2 > UAS-TNT larvae or for 60 s). The coordinates of the 11 points along the central spine and the outline of each larva were computed as described previously[47,48]. These 2D X-Y coordinates, structured as time series of irregular framerate, labeled by the instantaneous tracking time of the recording, comprise the raw datasets, which have been subsequently analyzed to derive all secondary metrics.

Analysis was performed in Python using the larvaworld behavioral analysis and simulation platform (https://pypi.org/project/larvaworld/). The 3-step analysis pipeline included preprocessing, computation of secondary angular, translational, and temporal metrics, and behavioral epoch detection to annotate strides, runs, and pauses, as described previously[68]. During preprocessing, the raw time series were adjusted to a 10 Hz constant framerate by interpolating them at a 0.1 s timestep. Noise reduction was achieved by applying a low-pass filter with a 2 Hz cut-off frequency, a threshold high enough not to alter the crawling-related dynamics around the dominant ~ 1.4 Hz crawling frequency.

For trajectory-based spatial metrics such as pathlength and dispersal, to avoid the cumulative effect of body micromovements, the position of the 9th point along the midline was used as a proxy for the larva's position, a relatively stable rear point unaffected by lateral and translational jitter. To correct for different larval sizes, any metric measured in absolute spatial units (m or mm) can be scaled to body length, measured in dimensionless body-length units. As the instantaneous body length of an individual larva fluctuates during crawling due to subsequent stretching and contraction, individual larva length is defined as the median of the midline length across time (total length of the line connecting all 11 midline points). A trajectory's path length is the cumulative displacement of the larva during the entire track. Dispersal is the instantaneous straight-line distance relative to its initial position. Track tortuosity was quantified by the straightness index (S.I.), computed by advancing a fixed time window (20 s in this study) along the track and calculating at each point the ratio of the dispersal to the actual distance traveled. This index, which varies from 0 (no movement) to 1 (straight line movement), can capture very well the complexity of the movement at various scales (set by the window time frame) throughout the track.

For the detection of peristaltic strides and crawl-pauses, the scaled crawling speed time series were used. To this end, the dominant crawling frequency for each track was extracted by applying a Fourier analysis, and its inverse was used as the expected duration of a peristaltic cycle. A stride was therefore defined as the epoch between two

local speed minima, that included a local maximum of at least 0.3 body-lengths/s and lasted between 0.7 and 1.5 times the expected cycle duration. A run was defined as an uninterrupted sequence of consecutive strides, and a crawl-pause as an epoch lacking any strides during which the scaled speed was constantly below the 0.3 body-lengths/s threshold.

## Stimulation during behavioral tracking

Air-puff was delivered as described previously[32,47,48] to the 25625 cm2 arena at a pressure of 1.1 MPa through a 3D-printed flare nozzle placed above the arena, with a 16 cm × 0.17 cm opening, connected through a tubing system to plant supplied compressed air. The strength of the airflow was controlled through a regulator downstream from the air amplifier and turned on and off with a solenoid valve (Parker Skinner 71215SN2GN00). Air-flow rates at 9 different positions in the arena were measured with a hot-wire anemometer to ensure consistent coverage of the arena across experimental days. The air-current relay was triggered through TTL pulses delivered by a Measurement Computing PCI-CTR05 5-channel, counter/timer board at the direction of the MWT. The onset and duration of the stimulus were also controlled through the MWT. Larvae were left to crawl freely on the agar plate for 60 s prior to stimulus delivery. Air-puff was delivered at the 60th second and applied for 30 s.

For optogenetic experiments, light was delivered using a custom-made 16 × 16 LED panel. The arena was also illuminated from below with IR light. Light (alone or with air puff) was triggered at the 60th second and lasted for 30 seconds. The light intensity was measured as the irradiance (mW/cm²) using a PM16-130 photometer (THORLABS). Irradiance was measured at 12 points across the arena and then averaged. The light intensity used for optogenetic activation experiments (red light, 617 nm) was 0.3 mW/cm².

## Behavior classification and analysis

Behaviors were detected thanks to a custom-made machine learning algorithm that was previously described[47]. Behaviors were defined as mutually exclusive actions. Larvae were tracked using MWT software, all the time series of the contours and the spine of individual larvae are obtained using Choreography. From these times series, some features are computed, the center of the larva, velocities, etc., all key features are presented in Masson et al., 2020[47]. Behavior classification consists of a hierarchical procedure that was trained separately based on a limited amount of manually annotated data. Here, we required a more detailed definition of behavior. Hence, we extended the hierarchy with another layer to separate some Bends and Hunches between different behaviors. Bends, which were defined as a large action category encompassing all behavioral dynamics involving bending of the larva body, were separated into Static Bend and Head Cast (see the description of each behavior below). We take all the Bends, Hunches, and Back-ups obtained by the first classification algorithm, to reclassify with new annotated data on new lines.

## Action definition

**Head cast**. Dynamic bends in which the head moves laterally from one side to the other, or sometimes from one side to the center. There are two exits from Head Cast, in both, the head moves strongly. In one, the barycenter does not move significantly, and the action at the end of the head cast is a run. In the other, the larva body moves more, and the end of the head cast is a turn (in which the larva transitions to a run with a curved trajectory).

**Static bend**. Low speed motion where the larva bends its body without its barycenter moving significantly. Static bends start with slow motion of the head and the angle between the segment between the center of mass and the head and the center of mass and the tail remains constant.

## New annotated data

Strong differences in phenotype led to features associated with larva actions being statistically too different from the original training dataset. Hence, new training data were needed to adapt the classifiers. The sets of Hunch, Bends, Backs, and Crawls were manually tagged from actions selected using the Masson et al., 2020 old behavior classification pipeline. A few numbers of tags are used for the model.

## New features

The fine-tuning procedure consisted on adding an extra layer to train on top of the main architecture. We combined features that were previously computed to provide classification within the main architecture with new features. Each characteristic is calculated for the time step we are examining and the three time steps before and after.

- The three velocities include the head velocity, the motion velocity, and the tail velocity, all normalized by the length of the larva. The motion velocity is the velocity of the center of mass return by the MWT software. Head and tail are the terminal points of the spine; The averaging along the spine curve and its derivative, $S = \frac{1}{2}(3\langle cos^2\theta \rangle - 1)$ with $cos\,\theta$ the scalar product between normalized vectors associated to a segment of the spine and the direction of the larva body.
- The shape factor $\lambda = \frac{\lambda_1 - \lambda_2}{\lambda_1 + \lambda_2}$ with $\lambda_i$ the eigenvalues of the mean covariance matrix of movement, which characterizes the shape of the larva and takes a value between 0 and 1.

We have also introduced new features:
- The ratio between the length of the head-center of mass and the tail-center of mass $\frac{||\overrightarrow{HG}||}{||\overrightarrow{TG}||}$ with $H, T$, and $G$ respectively coordinate points of the head, the tail, and the center of the mass.
- The projection of the head and tail velocity on the spine of the larva. If we note the velocity vector of the head $HV_h$ with H coordinate point of the head and $V_h$ the coordinate point at the end of the velocity vector, the projection point satisfies the basic relationship: $||V_h - V_h'|| = \min ||V_h - x||$ with $x \in \overrightarrow{r_{si}}$.
- The cosine of the angle between the vector of the head (tail) velocity and the first (last) segment of the larva. $cos(\theta) = \frac{\overrightarrow{r_{s1}} \cdot \overrightarrow{v_{head}}}{||r_{s1}|| \, ||\overrightarrow{v_{head}}||}$ with $\overrightarrow{v_{head}}$ the vector velocity of the head and $\overrightarrow{r_{s1}}$ the first vector of the spine.

All features are normalized by the length of the larva to ensure scale-free properties.

## New classifications

The fine-tuning procedure consisted on adding an extra random forest on the main classifier architecture, acting at each time point to re-classify actions that were classified as Bends, Hunches, and Backs. We conducted ten random forests on all Hunch and Back tags, along with a random selection of a thousand Bends, utilizing balanced weights. The predicted behavior represents the most probable outcome, with each random forest's confusion matrix exceeding 80% accuracy for each behavior.

We regularized anomalies of behavior lasting less than 3 time points, which is biologically unrealistic. To address actions spanning only two time steps, we introduced a preventive measure by adding five time steps before and after the behavior. In addition, to mitigate noisy results, we implemented smoothing. This involved excluding behaviors with fewer than three time steps, logically categorizing them as the behavior before or after (based on the length of the behavior and behaviors N + 2 and N-2). According to our knowledge, Hunch behavior initiates at the beginning of stimulation, around 60 s[32,42,47]. Larvae typically do not exhibit multiple Hunches. In cases where they

do, we classify the second Hunch as a Head Cast, a classification verified through ground truthing. We applied the same threshold as outlined in Masson et al., 2020 to the effective length change during the behavior. If a Hunch fails to surpass this threshold, the time window is assigned to the small behavior. The threshold is not the same depending on the line, some lines were slower or smaller than others (threshold between 0.6 and 0.3). The validation of these thresholds was performed through ground truthing, contributing to the enhancement of classification, particularly in cases where performance may be suboptimal for certain lines.

The distinction between Static Bends and Head Cast is determined by applying a threshold to the head velocity. If bending occurs over 'n' time steps, the motion velocity normalized by the length of 'P%' of those 'n' time steps must be below 'p' times the mean head velocity of the larva before the stimulation. The values of the two thresholds, 'P' and 'p', are line-specific, contingent upon the statistical characteristics of the velocity for that particular line; some lines are slower than others.

We computed the cumulative probabilities of actions (Stop, Hunch, Back-up) during five seconds after stimulus onset (as described in Masson et al., 2020), only in larvae that were tracked during the entire time window. For actions that occur during baseline locomotion and at high frequency (Crawl and Head Cast), we computed the mean probability over five seconds after stimulus onset and over three seconds starting one second after stimulus onset. We corrected the mean probabilities for these actions by the mean probability computed over twenty seconds of recording prior to stimulus onset. For optogenetic activation of the mechanosensory neurons, because the dynamic of the response was different to that of air-puff experiments, the time window used for computing the cumulative probability of Hunching was two seconds after the stimulus onset and that for bending probability was ten seconds after stimulus onset, by which time bending probability reached baseline levels in larvae fed on standard food. Transition probabilities were computed as the frequency of transition from one action to another over five seconds after stimulus onset, for larvae that were tracked throughout this entire duration.

## Unsupervised behavioral analysis

As an alternative approach to mapping tracking data onto a dictionary of discrete actions, we projected the same tracking data in a 25-dimensional space to represent behavior in an unsupervised and continuous fashion.

We specifically did so for the experimental conditions that suffered from poor action identification, namely the NPFR genotype and its control.

We generated the 25D behavioral readout using an autoencoder technique known as MaggotUBA[69], which takes 2-second time segments as inputs.

As a consequence, for each larva, 5 evenly-spaced time points were selected in the 60–65 s time window (stimulus onset at 60 s).

A 2-second time window was centered at each of these time points.

The spatial coordinates of the 5-point spine were collected within the time window, resampled at 10 Hz, and concatenated in a 200-element vector.

All such raw data vectors were projected into the common 25-dimensional space using a MaggotUBA encoder named 20230129 and publicly available as part of the LarvaTagger project[70,71].

The 25D data points were used to compare between feeding states.

For each pairwise comparison, the Maximum Mean Discrepancy[72] (MMD) was computed with a Gaussian kernel of width 2.

A permutation test was applied to generate surrogate MMD estimates and derive a p-value.

After inspecting a total of 45 pairwise comparisons, we show the 4 most relevant pairwise comparisons.

In particular, we only show the 60.5–62.5 s window, as it exhibits the strongest effect and leaves a short time interval after stimulus onset, which guarantees the stimulus onset took place just before, in spite of noisy timestamps from the tracking software

The p-values were Bonferroni-corrected for the actual number of comparisons (45).

To illustrate the number of data points and groups (Supplementary Fig. 9), a 2D representation of all the data points was generated using supervised UMAP[73] (Python package umap-learn, with parameter n_neighbors = 20).

The categorical information included genotype and feeding state.

As expected, all the groups can be easily separated.

However, interestingly, the various groups related to the control genotype, together with the fed NPFR larvae are closer neighbors than the sucrose-fed and starved NPFR larvae.

Note that supervised UMAP successfully separates all the experimental conditions due to its ability to leverage local neighborhood relationships. If translated to a supervised classification approach, the domains associated with each experimental condition in the 25D latent space would exhibit complex involucrated shapes, which would be considered a sign of overfitting. The resulting 2D arrangement of the different groups of points is still informative, in spite of the complexity of the mapping. In contrast, the MMD relies on simpler and more easily interpretable patterns, as it compares two groups of points using the moments of the respective distributions. Basically, and most likely, a significant MMD indicates a difference in mean and/or variance.

## Food intake quantification

In order to measure food intake, we quantified the amount of fluorescent food inside the digestive tract of larvae that were allowed to feed ad libitum for 15 min on fluorescent feeding media. To this end, Rhodamine B (Sigma R6626-25G) was diluted in different feeding media (water, yeast extract, sucrose solution, or normal food medium, see in media section) to a final concentration of 20 μmol/L (78 μmol/L for Fig. 1j). 0.8 ml L of each food medium was poured on top of a circular Whatman paper (Fisher Scientific, Cytiva 1001-045) placed into a petri dish. 10 larvae of similar size were placed into each petri dish. After 15 min of ad libitum feeding on the fluorescent medium, larvae were collected, rinsed in ethanol and in water, and immediately mounted under a coverslip for imaging. Intact larvae were imaged thanks to a fluorescent binocular Zeiss Discovery.V12. The surface of the digestive tract stained by the fluorescent dye was quantified for each larva thanks to custom-made Fiji and Matlab scripts. The scripts quantify the number of pixels whose intensity was above background, extracting the corresponding values of surface (area) and intensity. All area values were normalized by division by the average area of the control larvae fed on the standard food of each experimental day. The normalized area and the fluorescence intensities are multiplied to obtain the values to compare food intake between groups.

## Sucrose preference

To measure the preference of larvae between a sucrose-containing agar and a water-containing agar, we performed a place-preference assay. To this end, we prepared petri dishes filled with 0.3% agar diluted in water, divided each agar in two halves, and transferred one half into a new petri dish. Then, we filled the missing half in each Petri dish with 0.3% agar diluted in a 20% sucrose solution. Therefore, each petri dish finally contained one side with 20% sucrose agar and one side with agar only.

After cooling down, about 20 third instar larvae were put on the midline of a petri dish, and the dish was imaged every 30 s in a behavioral tracker. The number of larvae on each side was then counted over time, excluding the larvae touching the limit between the two

media. The preference index was then calculated as: PI = (Ns - Na)/Ntot, where Ns is the number of larvae on the sucrose side, Na is the number of larvae on the agar side, and Ntot the total number of larvae.

## Carbohydrate measurements

To measure carbohydrate concentrations inside larval hemolymph, we combined and adapted different methods already published[74,75]. Glucose measurements - Groups of 5 third instar larvae were rinsed in water and placed on a parafilm layer and their cuticle was cut with forceps. 2 µL of the bleeding hemolymph was collected from each group, and 1 µL of 0.05 g/L N-phenylthiourea diluted in PBS was added to avoid darkening of the samples. Samples were heat-inactivated by a 10 min incubation at 90 °C and centrifuged 10 min at 9500 x g. 1 µL of the supernatant was then mixed with 4 µL of glucose assay kit (Sigma GAHK20-1KT) and incubated for 1 h at 37 °C. Absorbance at 340 nm was measured against the blank thanks to a NanoDrop following the manufacturer's instructions. Hemolymph glucose concentration was finally calculated thanks to a standard curve of glucose concentration.

Trehalose measurements - Groups of 10 third instar larvae were rinsed in water and placed on a parafilm layer, and their cuticle was cut with forceps. 3 µL of the bleeding hemolymph was collected from each group, and 97 µL of 0.05 g/L N-phenylthiourea diluted in trehalase buffer (Tris pH 5.5, 5 mM, NaCl 137 mM, KCl 2.7 mM) was added to avoid darkening of the samples. Samples were heat-inactivated at 70 °C for 10 min, centrifuged for 10 min at 10,000 rpm, and 5 µL of the supernatant were either mixed to 5 µL of trehalase (Sigma T8778-1UN) diluted in trehalase buffer (described above) to a 500 dilution. Samples were incubated at 37 °C overnight. 1 µL of each sample was then mixed with 4 µL of glucose assay kit (Sigma GAHK20-1KT) and incubated for 1 h at 37 °C. Absorbance at 340 nm was measured against the appropriate blank thanks to a NanoDrop following the manufacturer's instructions. Trehalose concentration was finally calculated thanks to a standard curve of glucose and trehalose concentrations, and by additionally subtracting the concentration of glucose in the sample without trehalase.

## Histochemistry labeling

To determine the neurotransmitter identification in the interneurons, immuno-labeling was performed from the split lines or Gal4 lines crossed to UAS-myr::GFP, or LexA lines crossed to LexAop-myr::GFP. The VNC was dissected out from 3rd instar larvae and fixed with 4% PFA for 45 min at room temperature. After rinsing in PBS, ten minutes of permeabilization in PBS-T and two hours blocking in PBS-T-BSA 1%, the CNS preparations were incubated at 4 °C (one to three nights) in the first antibodies raised against neurotransmitter and GFP in PBS-T. Then they were incubated at 4 °C (one to two nights) in fluorophore-coupled secondary antibodies in PBS-T raised against the species of the first antibodies. After rinsing, the preparations were mounted in an anti-bleaching mounting medium (SlowFade Gold, ThermoFisher S36939) under a cover slip. The confocal images were captured with a Leica SP8 confocal laser microscope. Alexa Fluor 488 was excited with a laser light of 488 nm, Cy3 with a laser light of 561 nm, Alexa Fluor 647 with a light of 633 nm wavelength.

## Neuropeptide receptor characterization

In order to characterize the expression pattern of sNPFR and NPFR in the circuit, expression of two genetically encoded reporter proteins of two different colors was targeted to two different subsets of neurons. To this aim, a T2A Gal-4[52] or LexA (for sNPFR and NPFR, respectively) was used to express LexAop-jRGECO1a or UAS-GCaMP6s (for sNPFR and NPFR, respectively) under the control of the promoter of the gene coding for that receptor, thus tagging all neurons which express the receptor transcript. A second genetic driver (LexA for sNPFR and Gal4 for NPFR) was used to individually label target neurons of the circuit with a second reporter protein (UAS-GCaMP6s for sNPFR and LexAop-

jRGECO1a for NPFR). The VNC was then dissected and stained with antibodies as described in the previous section.

To the best of our knowledge, no clean or sparse LexA line exists to selectively target the Handle-b and Griddle-2 inhibitory interneuron. For sNPFR expression in Griddle-2, the LexA line L55C05 used targets many neurons in addition to Griddle-2. Griddle-2 could nevertheless be identified by comparing the cell body position and projections in the cross-section of the anterior abdominal segments of the CNS of the R55C05 LexA line and the sparse R55C05 GAL4 line that selectively labels Griddle-2 in the VNC (see Supplementary Fig. 7d).

For Handle-b, we used a L60E02-LexA line for which a neuron with a cell boy in the midline resembling Handle-b is part of a very dense expression pattern. In order to confirm that the candidate enron was indeed Handle-b, we used the selective split-Gal4 line GMR_SS00888 that specifically labels only Handle-b neurons. We expressed two reporters of different colors (LexAop-GCaMP6s and UAS-Chrimson-mCherry under the control of GMR_SS00888 and L60E02 in the same larva. The colocalization confirmed the line 60E02-LexA to target Handle-b (Supplementary Fig. 7e).

## In vivo imaging of intact larvae

Because opening the cuticle might affect the larval internal state, we developed a simple preparation for the imaging of intact larvae. For this purpose, third instar larvae were rinsed in water, and mounted between a 2 cm circular coverslip and a custom-made device that delivers mechanical stimulations in low melting point agarose 4% (melted in phosphate buffer saline), ventral side facing up. Larvae were gently squeezed in this position until the agar cooled down, so that the ventral nerve cord could be imaged through the cuticle.

All Gal4 and LexA drivers used for in vivo imaging are listed in the figure legends. The imaging plane was restricted to the location of the projections of neurons of interest, in particular when sparse lines that lacked specificity towards a unique neuronal type were used (R22E09).

Mechanical stimulations were generated by a waveform generator (Siglent sdg1032x) connected to a quick-mount extension actuator (Piezo Systems, Inc.), which was embedded in the sylgard-coated recording chamber (Sylgard Silicone Elastomer, WPI). The stimulation was set at 1000 Hz, with an intensity of 1 to 20 V applied to the actuator. The amplitude of the acceleration produced by the actuator was measured thanks to a triple-axis accelerometer (Sparkfun electronics ADXL313) connected to a RedBoard (Sparkfun electronics) and bound to the Sylgard surface thanks to high vacuum grease. Acceleration was $1.14\,m.s^{-2}$ at 20 V, and $0.61\,m.s^{-2}$ at 10 V. Mechanical stimulations were precisely triggered by the Leica SP8 software thanks to the Leica Live Data Mode and to a trigger box branched to the scanning head of the microscope. A typical stimulation experiment consisted in 5 s of recording without stimulation, then 5 s of stimulation, and 5 s of recording in the absence of stimulation.

For optogenetic activation during in vivo imaging, larvae were mounted in the dark, with the least intensity of light possible in the room, to avoid nonspecific activation of the targeted neurons. Optogenetic stimulation of CsChrimson was achieved by a 617 nm wavelength LED (Thorlabs, M617F2), controlled by an LED driver (Thorlabs, LEDD1B) connected to the waveform generator, and conveyed through a Ø 400 µm Core Patch Cable (Thorlabs) to the imaging field. Optogenetics stimulations were triggered at 50 Hz, 50% duty during 1 s, concomitantly to mechanical stimulations thanks to the waveform generator. Irradiance was measured at the level of the imaging field at 500 µW using a PM16-130 THORLABS photometer.

Imaging was achieved with a 1-photon or 2-photon scanning Leica SP8 microscope, at 200 Hz, with a resolution of 512 × 256 pixels or 512 × 190 pixels. The rate of acquisition was 1 frame/s or 2 frames/s, depending on the experiment. For Basin-2 recordings, the stimulation was repeated 5 times with resting intervals of 60 s in order to calculate a frequency of response. Recordings where the dF/F averaged over the

whole stimulus duration did not exceed 10% were considered as failed responses. Optogenetic experiments were conducted with 2-photon imaging.

When the projections of the neurons were not visible before stimulation (imaging of Handle-b upon NPFR knockdown), we used resonance scanning with 10 line accumulation and 6 frame averaging to increase the signal. This resulted in one image being taken each second and in the dF/F being lower than usual. With these settings, one recording was acquired per larva.

Neuronal processes were imaged in the VNC at the axonal level, and fluorescence intensity was measured by manually drawing a region of interest (ROI) in the relevant areas using custom Fiji macros. Data were further analyzed using customized MATLAB scripts. F0 was defined as the mean fluorescence in the ROI during baseline recording, in the absence of mechanical stimulus or optogenetic activation. ΔF/F0 was defined at each time step t in the ROI as: $\Delta F/F0 = (F(t) - F0)/F0$.

For recording the baseline activity level of NPF neurons, one frame was recorded each second for 20 s, and, for each larva, the frame showing the most intense fluorescence in the neuronal projections was used to evaluate its raw fluorescence level.

For imaging neurons in different feeding conditions, the effect of food treatment on the expression level of fluorescent reporter proteins was assessed by expressing GFP in the chordotonal neurons. Fluorescence was measured after exposing larvae to different food treatments.

### Imaging in filet preparation

A filet preparation was used for recording baseline activity in DM-NPF neurons upon TTX applications. The filet preparation was performed as previously described in Jovanic et al, 2016. Briefly, a longitudinal incision was made along the larva's midline, carefully removing all organs, except for the nervous system. The cuticle was carefully stretched and secured at the corners for optimal exposure of the VNC. Dissections were carried out in Artificial Hemolymph-like (AHL) solution, composed of 103 mM NaCl, 3 mM KCl, 5 mM HEPES, 1.5 mM $CaCl_2$, 4 mM $MgCl_2{\cdot}6H_2O$, 26 mM $NaHCO_3$, 1 mM $NaH_2PO_4{\cdot}H_2O$, 10 mM trehalose, 7 mM sucrose, and 10 mM glucose (Boto, T. and Tomchik, S.M., 2024).

To determine whether NPF neurons respond to glucose and whether this response is mediated by synaptic communication, calcium levels in NPF axonal projections were measured following the application of 10.5% glucose (osmolarity-matched to 20% sucrose) in AHL, both before and after synaptic blockade with tetrodotoxin (TTX). Neuronal viability throughout the experiment was confirmed by assessing calcium responses to high $K^+$ saline at the end of the experiment. NPF axonal processes in the ventral nerve cord (VNC) were sequentially imaged in the following solutions, replacing the previous solution each time: (1) AHL, (2) 100 μL of 10.5% glucose in AHL, (3) 100 μL of 20 μM TTX in AHL (incubated for 5 minutes to allow TTX action), (4) 100 μL of 10.5% glucose in AHL, and (5) 100 μL of high-$K^+$ (100 mM) saline (Boto, T. and Tomchik, S.M., 2024).

Imaging was performed using two-photon scanning on a Leica SP8 microscope at 400 Hz with a resolution of 512 × 512 pixels. Z-sections were acquired every 2 μm. Equal numbers of larvae were imaged across the Fed, Sucrose-Fed, and Starved conditions within each experimental day. Fluorescence intensity was quantified by manually defining regions of interest (ROIs) in the relevant areas on the maximum Z-projected image using custom Fiji macros. Background fluorescence in the ventral nerve cord (VNC) was subtracted for each larva.

### Statistical analysis

**Locomotion analysis.** For all boxplots and histograms, pairwise Mann-Whitney tests were used to evaluate significant differences between larva groups, with Bonferroni correction for multiple comparisons.

Significance was illustrated according to the p-value by asterisks in histograms (*:< 0.05, **:< 0.01, ***:< 0.001, ****:< 0.0001) and by pairs of colored semicircles in boxplots, the left always corresponds to the group with the highest mean value. Non-significant tests were omitted for visual clarity.

### Behavior probabilities

Behavioral probabilities were computed during the first five seconds of stimulation. Chi² tests were used for statistical comparison.

To assess the effects of different states/neuronal manipulations on Head Casting in response to air-puff, we calculated an estimator designed to identify the emergence of behaviors at the population scale. We aim to determine the probability induced by the stimulation, so we need to subtract the stationary probability without the stimulus. We calculate the probability of larvae Bending 5 s after the stimulus, on tracking larvae throughout this time window (denoted as $p_A$). For the probability of Head Casting ing before the stimulus, we consider all larvae tracked continuously for 20 seconds prior to the stimulus, between 30 and 50 seconds (denoted as $p_B$). A probability is defined as $p_k = N_k/N_{k,all}$ with $k \in \{A, B\}$ and $N_{k,all}$ the total number of larva taking to compute probabilities. In order to quantify the effect of the stimulus we defined $\chi = p_A - p_B$ as the difference in the ratio after and before the stimulus.

In order to compare the test line our estimator was defined as $\Theta(p, q) = \chi(p) - \chi(q)$ with $p$ and $q$ the ratios of the lines and the control respectively. $\Theta(p, q)$ takes value in [-1, 1]. The null hypothesis is $\Theta(p, q) = 0$, if there are no differences between the line tested and the control. Positive or negative values indicate an effect of neuron silencing when compared to the control.

We use numerical simulations to conduct a statistical test where $\{p_k, q_k\}$ are generated from a hypergeometric distribution, $X \sim Hypergeometric(N_{k,all}, N_k, 1)$. We perform $N_{sim} = 10^3$ repetitions, computing $\Theta(p, q)$ each time. The $p$-value is determined by the number of instances when the hypothesis is not verified, divided by the total number of repetitions, (pseudo-code in Jovanic et al., 2016[32]).

Note that this estimator, $\Theta(p, q)$, has the advantage of being able to detect the non-synchronous emergence of a behavior at a population scale. For example, Head Casting can either emerge as an immediate response to the puff or as the second response after Hunching. The statistics of the start time of Head Casting is thus widely distributed at the population scale. Time evolution of the instantaneous ratio of larva performing Head Casting ing would not exhibit a strong increase after stimuli because larvae are not all going to Head Casting immediately after stimuli. $\Theta(p, q)$ by accumulating events during a time window allows efficient detection of a behavior even if it is widely distributed in time.

Generalized likelihood ratio tests were used for statistical analysis of behavioral transition probabilities. An in depth description of this test can be found in Masson et al.[47]. In brief, the test statistic is computed as $z = -2log((B(\pi^k, N_m, n_m^k)B(\pi^k, N_0, n_0^k))/(B(\pi_m^k, N_m, n_m^k)B(\pi_0^k, N_0, n_0^k))$ with $\pi = \frac{n_m^k + n_0^k}{N_m + N_0}$ and $p = \chi^2(z, df = 1)$ as well as a chi² test.

We used short time windows (2 s) to assess the transition probabilities that happen immediately upon stimulus onset. For experiments where the number of transitions was low (< 100) during the short time window, we computed transition probabilities in the first 10 s of stimulation to make more robust comparisons between conditions.

### Calcium imaging

All data in line plots are presented as mean ± SEM. Violin plots show the first and third quartiles, the average of all recordings as a white line, and the median as a white dot. Comparisons of the data series between the two conditions were achieved by a two-tailed unpaired t-test. Comparisons between more than two distinct groups were made using

a one-way ANOVA test, followed by Bonferroni pairwise comparisons between the experimental groups and their controls.

## Mathematical model

We reproduced and extended the rate-based system model of the circuit that was published in a previous publication[32]. The circuit is described as a rate model with a connection matrix derived from the larva connectome. Each neuron population (mechano-ch, iLNa, iLNb, fbLN-Ha, and fLN-Hb) was modeled by a single node (Fig. 3h-p). The dynamics read:

$$\tau \frac{dr}{dt} = -V_0 - r + s + i + k_{ex}(r - r^{max})\left(\underline{A}^{ex}r - \underline{A}^{in}r\right)$$

with $\tau$ representing the vector of the characteristic time constants of the neurons, $r$ the rate vector, $V_0$ the threshold vector, $s$ the sensory stimulus input vector, $i$ the vector of inputs from other brain regions, $k_{ex}$ a sensitivity factor to overall input, $r^{max}$ the maximal rate vector, $\underline{A}^{ex}$ and $\underline{A}^{in}$ respectively the excitatory and inhibitory coupling matrices. Vector multiplication denotes element-wise multiplication, also called the Hadamard product.

Values in $\underline{A}^{in}$ and $\underline{A}^{ex}$ were directly extracted from synaptic counts (see Jovanic et al.)[32].

In order to represent the variety of stimuli larvae are subjected to, and thus the variety of behavior they elicit, we follow the approach in Jovanic et al, 2016 and vary the connection strength between Mch and iLNa populations. In the original paper, the connection strength between Mch and iLNb populations is also varied. We decided to fix the value of this connection strength at 2 in order to reduce the number of parameters to explore.

We used the *solve_ivp* routine of the integration package from SciPy, which internally calls the LSODA solver, able to switch between the Adams method and the BDF method, based on the stiffness of the equation. We used relative and absolute tolerances of $10^{-3}$. In addition, the solution vector is constrained to stay positive. This is obtained by replacing r by max(r, 0) and the components of r' by those of max(r', 0) whenever the corresponding component of r is smaller than $10^{-9}$.

The behavior is defined based on the steady-state rates of populations B1 and B2. We used a k-means clustering with $k = 2$ to separate the values of the output neurons for an ensemble of simulations corresponding to connection strengths spanning [0.5, 1.5] between MCh and iLNa, and [1.5, 2.5] between iLNa and iLNb. The output activations cluster strongly in a coactive state (rate(B1) $\gg$ 0, rate(B2) $\gg$ 0) corresponding to Head Casts, and a monoactive state (rate(B1) $\gg$ 0, rate(B2) $\simeq$ 0) corresponding to Hunches. The two behaviors can also be distinguished by the single scalar rate(B2)/rate(B1), which is large for Head Casts and small for Hunches. This ratio is sometimes plotted instead of the discrete category.

We explore two models for neuromodulation, which modify the behavioral output of the network without altering its connectivity.

The first model hypothesizes that in the sucrose state, Hb receives an additional input current, modeled by a nonzero entry to the vector $i$. We show that as this input current increases, the range of stimuli evoking Head Casts increases, allowing us to claim that an additional input current to Hb increases the likelihood of Head Cast. We further fix the value of the input current to 10 a.u. in the sucrose state and 0 a.u. in the fed state, to perform silencing analyses.

The second model hypothesizes that in the sucrose state, the maximum rate for the iLNa neuron population is decreased, representing a saturation of the response to external stimuli. We show that as the $r_{max}$ parameter for iLNa decreases, the range of stimuli evoking Head Casts s increases, consistent with experimental observations and once again despite the use of arbitrary units. For silencing analyses, we define the sucrose state as $r_{max} = 18$ a.u. and the fed state as $r_{max} = 20$ a.u.

Finally, we also consider a model combining both hypotheses. In this model, every combination of a decrease in $r_{max}$ for iLNa and an increase in input current to Hb results in more Head Casts. For the silencing analyses, we fix the values of those parameters. In the combined model, we define the sucrose state by setting each parameter to the value defining the sucrose state in the single-hypothesis models.

We provide here the list of parameters used in the simulations.

| $k_{ex}$ | 2.5 |
|---|---|
| $i$ | [0, 0, 0, 0, 0, 0, 0] in fed state<br>[0, 10, 0, 0, 0, 0, 0] in sucrose state for the combined model |
| $r_{max}$ | [20, 20, 20, 20, 20, 20, 20] in fed state<br>[20, 20, 20, 20, 20, 20, 18] in sucrose state for the combined model |
| $V_0$ | [0, 20, 20, 20, 20, 20, 20] |
| $\tau$ | [1, 35, 35, 35, 35, 35, 35] |

The excitatory and inhibitory matrices read

$$A_{ex} = \begin{bmatrix} 0 & 0 & 0 & 0 & 0 & 0 & 0 \\ 1.5 & 0 & 0 & 0 & 0 & 0 & 0 \\ 0.75 & 0 & 0 & 0 & 0 & 0 & 0 \\ 2 & 0 & 0 & 0 & 0 & 0 & 0 \\ W_{iLNa} & 0 & 0 & 0 & 0 & 0 & 0 \\ 0 & 0.2 & 0.2 & 0 & 0 & 0 & 0 \\ 0.4 & 0 & 0.5 & 0 & 0 & 0 & 0 \end{bmatrix}$$

$$A_{in} = \begin{bmatrix} 0 & 0 & 0 & 0 & 0 & 0 & 0 \\ 0 & 0 & 0 & 1.7648 & 1.3841 & 0 & 0 \\ 0 & 0 & 0 & 1 & 5.9167 & 0 & 0 \\ 0 & 0 & 0 & 0 & 3.3744 & 1.6659 & 2.191 \\ 0 & 0 & 0 & 2.7133 & 0 & 1.1010 & 3.3031 \\ 0 & 0 & 0 & 1.8411 & 1.1158 & 0 & 0 \\ 0 & 0 & 0 & 1.7331 & 2.2145 & 0 & 0 \end{bmatrix}$$

## Reporting summary

Further information on research design is available in the Nature Portfolio Reporting Summary linked to this article.

## Data availability

All data supporting the findings of this study are available within the paper, its Supplementary Information, as source data, and/or within the repository: https://doi.org/10.5281/zenodo.10959533[76]. Source data are provided in this paper.

## Code availability

Scripts used in this paper are available at:https://archive.softwareheritage.org/swh:1:dir:3f06d6d731741c4beebe901f5ef0b909c8b0b6bd;origin= https://gitlab.pasteur.fr/flaurent/chloestaggers/;visit=swh:1:snp:7480ee779daabf5b5ac887096643587949adaf43;anchor=swh:1:rev:913a5bc94bab9f2ebee578ef624c92d2e46e0c19.

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

## Acknowledgements

This work was supported by ANR PIA funding: ANR-20-IDEES-0002 (T.J.), Agence Nationale de la Recherche: ANR-17-CE37-0019-01 (T.J.), ANR-NEUROMOD (ANR-22-CE37-0027) (T.J.), ANR-21-NEUC-0002 (T.J.) in the context of the CRCNS collaboration grant with National Science Foundation CRCNS (2113179) and Department of Energy (SC0021922) (to Brian H Smith), Fédération pour la recherche sur le cerveau (FRC) (T.J.), Fondation pour la Recherche Médicale: Équipe FRM EQU202303016317, Fondation des Treilles (E.T.), D.M. received a PhD fellowship from the Paris-Saclay University; Tramway, ANR-17-CE23-0016 (J.B.M.), the inception Project PIA/ANR-16-CONV-0005,OG (J.B.M.), Investissement d'avenir program under the management of ANR, ANR-19-P3iA-0001 (PRAIRIE 3IA Institute (J.B.M.), German Federal Ministry of Education and Research (BMBF DrosoExpect, 01GQ2103A, M.N.), Ministry of Culture and Science of the State of Northrhine Westphalia (iBehave, Netzwerke 2021, M.N.). P.S. received a PhD stipendship from the German Research Foundation (DFG-RTG 1960, 233886668, M.N.). The funders had no role in study design, data collection and analysis, decision to publish, or preparation of the manuscript. We thank Marta Zlatic for sharing a Split Gal4 line for DM-NPF neurons.

## Author contributions

E.T. behavioral and physiology experiments and analysis, Calcium-imaging experiments and analysis; Writing: original draft, figures; D.M. behavioral experiments and analysis, Calcium-imaging experiments and analysis, immunohistochemistry experiments; Writing: edits and revisions, figures; A.B. modeling; A.P. calcium imaging and food quantification experiments and analysis P.S. Data analysis; C.B. behavioral classification and analysis, statistical analysis; Writing, methods; F.V. behavioral experiments; V.S. behavioral and physiology experiments; P.A. behavioral experiments, A.H. behavioral and immunohistochemistry experiments S.A. immunohistochemistry experiments; F.L. behavioral classification and analysis M.N. supervision, funding acquisition; J.B. Methodology, supervision, funding acquisition; Writing- edits and revision; T.J. Conceptualization, analysis, supervision, funding acquisition and project administration; Writing: original draft, figures, edits and revision. These authors contributed equally as co-first authors: D.M., E.T. These authors contributed equally as co-second authors: A.B., A.P.

## Competing interests

The authors declare no competing interests.
