## [Transparent Peer Review file · Nature Communications]

Feeding-state dependent neuropeptidergic modulation of reciprocally interconnected inhibitory neurons biases sensorimotor decisions in *Drosophila*

Corresponding Author: Dr Tihana Jovanic

Version 0:

Reviewer comments:

Reviewer #1

(Remarks to the Author)

The study by Tredern et al. explores how physiological states like hunger affect behavioral decisions, specifically focusing on the neural circuits involved in such state-dependent decisions using *Drosophila* larvae as the experimental model. The authors aim to understand how hunger and thirst influence non-feeding related behaviors such as threat avoidance, which involves choosing between different actions like "Hunching" (protective) and "Head Casting" (exploratory). The methods involved combining comprehensive circuit mapping using electron microscopy (EM) with targeted manipulations of identified neurons. Behavioral responses to an aversive air-puff stimulus were monitored using automated tracking and machine-learning-based classification to analyze the probability of different actions. The authors show that short-term food deprivation leads to increased locomotion and exploratory behaviors. Starved larvae and larvae fed only on sucrose exhibited more Head Casting and less Hunching compared to normally fed larvae. This behavioral shift was attributed to differential modulation of two reciprocally connected inhibitory neurons: one promoting Hunching and the other inhibiting it. Neuropeptides NPF and sNPF were identified as key modulators affecting the activity of these inhibitory neurons, thereby biasing the larvae's response towards more exploratory actions in a feeding state-dependent manner. Together, this work provides insights into how internal states influence behavior by altering neural circuit activity, demonstrating that hunger and thirst can modulate non-feeding related behaviors through specific neural mechanisms.

Overall, I find the subject of the manuscript to be interesting and relevant, and the paper is for the most part well written. The hypothesis that internal feeding states affect non-feeding behavioral programs via release of neuromodulatory hormones is elegant and attractive. However, the study lacks further experimentation and validation in certain parts of the study, which together prohibits me from recommending publication at this point. It should be further noted that although the work on larval behavior and the associated computational work seems very carefully planned and executed, this is not within my area of expertise and I have therefore not made detailed comments on these parts of the manuscript. I have outlined my concerns about the manuscript below.

Major comments:

1. In Fig. S1 the authors report that rehydration of larvae after feeding only on 20% sucrose restores their behavioral outputs to levels similar to that of controls. These data imply that the observed behavioral changes reported are, at least partly, induced by osmotic disturbances rather than by an increased motivation to locate nutrients as the authors also later argue (L155-161). Indeed, 20% sucrose represents a major osmotic challenge to the animals. This raises two specific points: 1) it remains unresolved whether the increase in yeast consumption is due to an increase in protein-specific appetite (A2) or simply a result of the animals attempting to restore osmotic balance by consuming a more dilute medium (i.e. high water). 2) the authors should also make further attempts to separate osmotic and nutrition inputs on behavior by considering exposing animals to the same osmotic challenge using non-nutritive osmolytes (i.e. inorganic salts and/or sucralose to produce a medium with the same osmotic pressure as that of 20% sucrose) and test if they still observe the same reversion of behavioral outputs?

2. The method employed for quantifying food intake I am not familiar with, but I see some inherent limitations of this

approach. Given that the authors only quantify the surface area of the gut occupied by the food (i.e. pixel number above background fluorescence) they don't collect information on pixel intensity, which would be equally important of assessing the amount of food intake. Further, they employ an epifluorescent microscope to collect pictures in 2D this means that they won't capture areas of the gut that overlap with each other, which would lead to wrongful estimations. Finally, the approach is also very sensitive to changes in larval size (e.g. large larvae bigger guts) and given the signal is not normalized to total gut size this can create large variation. In general, all assays for estimating food intake have limitations, which means that you should always use multiple approaches to assess food intake. In my experience, only when several assays point in the same direction can you be sure of what is happening. I would strongly advise the authors to supplement their food intake experiments with another approach, e.g. the blue dye assay that is used successfully by many labs, to independently test food consumption.

3. The authors propose that the NPF neurons are activated upon sugar-only feeding as well as on starvation (water only) relative to the normally fed larvae. This would imply that these neurons are able to integrated information, either directly or indirectly, on internal changes in sugar and water levels, as also shown for other neurons, albeit in the adult brain (1, 2). How might this be achieved mechanistically? Although the exact molecular players can be hard to identify, whether this is sensed autonomously or non-autonomously could be interesting to determine and could be tested by silencing synaptic input using TTX. This might help to connect and integrate this work with future studies on central nutrient sensing in *Drosophila*.

Minor comments:

L122 "Crawling" is written with capital c. Is this on purpose?

L151-152 If there is not a significant difference then you cannot claim that glucose levels are lower between fed and starved animals. Similar arguments are made throughout the manuscript (e.g. L255). Please adjust accordingly.

L419-423 I am curious that how the authors confirmed that the neurons were indeed NPF+ neurons? Was this solely done based similar morphology or was this done using the publicly available larval brain connectome?

Figures.

Fig. 1: Mentioning figure panels in the text in a non-chronological order seems counter-intuitive and is likely to be subject to journal guidelines. Consider organizing data such that they are mentioned in chronological order, i.e. panel 1A before 1B etc.

Fig. 3. There is a black line near "hunch" in panel E that needs to be removed.

Fig1S: Here there is a pronounced difference in the average Max dispersion distance of the starved larvae from the 5h treatment (ca 26 m distance) and the 5h treatment + refeeding (ca 15 m distance). What is the reason for this relatively big difference between identical exposures? In general, using similar scale sizes for the same readouts across experiments would make comparisons much easier. Similarly, what might be the reasons for the markedly lower responses across all locomotor outputs in 18h starved vs 5h starved animals? Are these observations linked with nutrient depletion of the animals?

Fig2S: in panel C the UA unit is not known to me. Is it meant a.u. as in arbitrary units?

1. Koyama T, Terhzaz S, Naseem MT, Nagy S, Rewitz K, Dow JAT, Davies SA, and Halberg KV. A nutrient-responsive hormonal circuit mediates an inter-tissue program regulating metabolic homeostasis in adult *Drosophila*. *Nature Communications* 12: 5178, 2021.

2. Jourjine N, Mullaney BC, Mann K, and Scott K. Coupled Sensing of Hunger and Thirst Signals Balances Sugar and Water Consumption. *Cell* 166: 855-866, 2016.

Reviewer #2

(Remarks to the Author)

This manuscript describes the neural basis of how the internal state influences the neural circuit controlling the responses to aversive mechanical stimuli in *Drosophila* larvae. The authors begin their investigation by examining how hunger affects the bias between two competing behaviors in response to air puff: hunching and bending. They reveal that the activities of groups of feedback or feedforward inhibitory neurons are influenced by the feeding state of the animal. Although the connectivity map of neurons underlying these behaviors is known, the authors discovered the nodes in this circuit that are subjected to modulation upon changes in the internal state. The authors discovered that a group of neuropeptide F (NPF) neurons is more active when the animal is hungry and modulates the activity of one of the feedback inhibitory neurons in the circuit. They also reveal that another neuropeptide, sNPF, inhibits the activity of these feedback inhibitory neurons, while it facilitates the activity of the feedforward neurons. In the era of connectomes, this study highlights the significance of neuromodulation in rendering neural circuits dynamic and adaptive to the context.

Overall, the study described in the manuscript is coherent and is a great follow-up to Jovanic and Schneider-Mizell et al. (2016) and Masson et al. (2020) studies. It is highly relevant to the field and will be of interest for the broad readership of *Nature Communications*.

Below are my notes and suggestions. I believe that a few additional experiments, some edits to the figures and the text, and some additional points of discussion would strengthen the work and make the reporting of the findings more concrete and clearer.

Major points:

1. In line 489 in the text, the authors state: "This increase in activity then biases the sensorimotor decisions towards less Hunching and more Head Casting." This statement should be supplemented further with behavioral experiments by knocking-down NPFR in Hb neurons and measuring hunching/ bending.
2. Figure 6F: A more proper no RNAi control for this experiment would be Gal4> UAS-GCamP; empty RNAi vector (a non-targeting RNAi, i.e., the valium vector for the TRIP collection).
3. Inferring from the sNPF-R knock-down experiments in Figure 7, sNPF plays a satiety signal. This notion should be tested with sNPF or sNPF-R knockdown experiments in starved animals to see whether they exhibit fed-like behaviors (i.e., hunch and bends).
4. The authors should present the individual data point throughout instead of simply using bar graphs.
5. In Figure 1 panel B1, the authors show that glycemia is increased in sucrose-fed animals and it does not change in starved animals compared to the control fed animals. Yet, in the text, the authors conclude that behavioral changes they observe is 'due to lack of nutrients and lower energy levels' (lines 163-165). This interpretation should be revised. On that note, how do the authors explain their findings that sucrose feeding and starving the animals produce the same behavioral outcome although they affect glycemia differently? This should be addressed in the discussion.
6. In Figure 3 and Extended Figure 3, Basin 2 neurons show differential activity in sucrose-fed and starved states: there is an increase in activity in sucrose-fed animals and decrease in starved (although not significant). How do the authors explain that these two states lead to the same behavior (i.e., increased bending in Figure 1 E1)? This should be discussed in discussion.
7. Comparing Figures 4 A-B to Extended Figures 4 A-B. How do the authors explain that the differences in calcium activity are more prominent in sucrose fed animals compared to starved animals? Given that both Griddle-2 and Hande-B neurons are modulated by the feeding state of the animal, the expectation would be that total starvation would have greater effect on the activity of these neurons compared to the sucrose only-feeding. This should be discussed in the discussion.

Minor points:

1. To increase clarity and make the paper more fluent and easier to read, the figures should be put in the order of their mentions in the text.
2. Extended figures 1B-D median lines for controls are not visible.
3. Figure 1E- the error bars are missing.
4. Lines 330-332: "The increase in Griddle-2 responses upon Handle-b inactivation in larvae fed on sucrose did not reach the levels of Griddle-2 responses upon inactivation in larvae fed on standard food (Fig.4C2-C3, Extended Data Fig. 4E)." This is not clear from the figures mentioned. Lack of significance should be indicated in Fig. 4C3.
5. Line 375: "As expected, silencing Griddle-2 abolished the difference in Basin-2 responses in the two States." It is unclear to which figure this text refers. This should be specified. Assuming that it is Figure 5C, the lack of significance should be pointed out.
6. Extended Figure 3C: Basin 1 activity-sucrose-fed trace should be red.
7. Figure 6A the EM image: what do the lines in different colors show? This should be addressed in the figure legends.

Reviewer #3

(Remarks to the Author)

Summary

In Tredern et al., the authors address important questions in neuroscience: how does the internal state of an animal modify behavior and, on a mechanistic level, how are neural circuits flexibly modified by the state to support these changes. To address this, they investigate the influence of hunger on the behavior and neurophysiology of *Drosophila* larvae, focusing specifically on modulation within a circumscribed neural circuit involved in escape decisions upon mechanosensory stimulation. This circuit was previously identified by members of the same group as the current paper, and the current paper extends this previous study in an interesting direction. While I find the overarching theme of the work exciting and the general strategy and organization of the manuscript sensible, I found multiple aspects of the results and/or the authors'

interpretations difficult to follow, and generally found the manuscript lacking concision. I clarify these and other issues below.

Major concerns:

1) The identification of “head casting” as an exploratory action seems an important detail of the behavioral description, but it seems that the distinction between this behavior and bending is not so clear throughout. At times, it seems the two terms are used interchangeably, which is confusing (e.g. line 280 says “bend” whereas line 296 says “head casting”). This becomes more critical when describing the circuit, where the focus becomes on the influence on basin-2 activity, which promotes bending. Is there any evidence that basin-2 activity is also related to “head casting”? If not, it's not so clear to me why the authors focus on “head casting” behaviorally. Related to this, why do the authors not also include “static bends” in their behavioral transition plots (e.g. in Figure 1E, right side)?

2) It is not clear why the authors present multiple types of hunger states in Figure 1 (sucrose-fed and starved) only to later primarily focus on the sucrose-fed state. As a reader, it is unclear why one state was favored over the other in later figures. The wording describing the Figure 1 results, for example, lines 139-148, is not straightforward, and it is a bit frustrating if the distinction between the two states is not clearly useful for later sections. In figure 2, for example, the same experiments were performed in both hunger states, but the starved data is buried in a supplement which makes it difficult to directly compare. By the time we get to figure 3, the starved state is no longer addressed. Are the differences observed between the two states relevant for the major findings of the paper? If so, how?

3) The modelling data presented in Figures 3 and 4 is poorly explained, both in the results section as well as in the figure legends. As a reader, I was not able to grasp their significance, and many of the plots contain multiple features that are simply not discussed anywhere. For example, in Figure 3C2, D2, E2, what is $wILNa$? What are the different lines? What is interesting or significant about the differences between the dotted (sucrose) versus the solid (fed) lines? In 3C3, D3, what is “affinity of input signal to $iLNa$ ”? This is not explained anywhere. Lines 290-292 mention “the sucrose state was modeled in one version of the model as...” but how does this translate to what is being shown in for example Figure 3C3, D3? My guess was at the lower r_{max} in figure 3C3 and the higher input current in 3D3, but I just wasn't sure I was reading this plots properly. I have the same issue with the model data in Figure 4C. These data are simply not explained in the text sufficiently for a reader to understand. Moreover, the specific models explored in Figure 3 seem rather contrived, basically matching the physiological results that follow in the subsequent figures. Some of the findings of the modeling also seem a bit trivial given the basic circuit (e.g. the silencing of Handle-b neurons described in lines 365-372). Are there alternative models that should be considered? As is, the models feel like throw-ins that do not add clarity or insight to the story, but rather force the reader to figure out how to interpret them on their own.

4) There are multiple aspects of the results where the authors' interpretation is unclear and/or hand-wavy. These interpretations sound speculative and do not add to the paper, but rather are clumsy to read through and distracting. For example, lines 267-271, 332-336, 347-350, 389-399, 408-411, 441-449, (sounds list-like), 467-471 (hard to follow/not helpful). Related to lines 389-399, the result that silencing Griddle-2 decreased Basin-2 responses when sugar fed struck me as really odd – the authors make note of this (lines 378-379), but I simply could not follow their subsequent interpretation.

5) It seems that the authors should instead do a Handle-b activation experiment instead of silencing to more explicitly test the hypothesis presented in lines 456-460 (related to figures 6B, D)? As is, it's not clear why the Handle-b silencing experiments (lines 451-460) were not done in a sucrose-fed state, where Handle-b activity was shown to be high.

6) The NPF-R staining shown in Figure 6E is not convincing. Can this be somehow quantified? Alternatively, could an intersectional strategy be used that includes $LexAop$ -flp and $UAS>stop>GFP$?

Other concerns:

1) The figures are not organized in line with the text, and this makes things difficult to follow. For example, in Figure 1, panel D is discussed before panel A in the text. Similarly, in Figure 5, panel B is discussed before panel A. This may occur elsewhere, but these two examples caught my attention.

2) Many of the panels also have multiple sub-panels that do not have unique identifiers. This makes it difficult to read the text and know which specific sub-panel is being referenced.

3) The axes on some plots are too zoomed out to discern the data. In Figure 1D for example, the axes make it difficult to assess the different groups. This also makes it hard to discern the RNAi results described in Figure 6C.

4) Why was “scaled speed” selected as the exemplar phenotype for the RNAi results in Figure 6C? Why not one of the others?

5) Comparing the inactivation data between fed and sucrose fed seems the major point of figure 3C3, but statistics were not performed between these two groups (the magenta outlined plots).

6) I find the control experiment described in lines 221-224 unorthodox. Is there a reference for this approach?

7) I am also not so familiar of making comparisons of GCamp raw fluorescence across animals, as was done in Figure 6B (lines 430-435). Is there a reference for this approach?

8) When introducing the behavioral classification analysis, it could be helpful to mention whether these are supervised or unsupervised algorithms (lines 188-196). The methods describing the "Behavioral classification analysis" is not very concise and difficult to read. The part about "static bends" versus "head casting" should perhaps be detailed in a supplementary figure as the text alone is not clear (lines 857-861).

9) Is it possible for the violin plots (e.g. Figure 2B) to include indications of the individual data points? I have seen this done with horizontal lines. This could make the interpretation of these plots, and the number of animals, easier to decipher.

10) When introducing the mechanical stimulus (starting on line 258), it's not clear that a range was used as the results only mention 5V, but the figure shows a range. Also, the text in line 260 seems it should reference figure 3B as well.

11) The results paragraph on lines 198-204 should mention that a 5s stimulus was used.

12) What is the significance of the statement on lines 203-204?

13) "Sucrose fed" is sometimes referred to as "protein-deprived" (e.g. in Figure 1C, Figure 5B) and this was quite confusing. The authors need to be consistent with this to avoid confusion.

14) Typo in Figure 1 legend: first five seconds "uoin" stimulus onset

15) Figure 2C doesn't mention which stimulus intensity is being shown (neither in the figure nor in the legend).

Reviewer #4

(Remarks to the Author)

Version 1:

Reviewer comments:

Reviewer #1

(Remarks to the Author)

The authors have, within reason, addressed all comments and suggestions and I therefore have no further points to raise with this manuscript.

Reviewer #2

(Remarks to the Author)

I am overall satisfied with the authors' revisions and would recommend accepting the manuscript for publication. That said, I have a few very minor suggestions for text edits that in my opinion would enhance the argument and clarity.

1. The new text added to the second paragraph of the Results section needs a better transition. As it is standing now it is abrupt and reads a bit of a non-sequitur. A transition sentence that explains the rationale (i.e., checking the possibility of the effect of stress due to the osmolarity of 20% sucrose).

2. The text discussing Supplementary Fig. 9 on page 11 is cursory. The data supports the arguments, and a bit more detailed description would help.

3. In the new text in the Discussion section (page 13) the term "lack of water" is awkward. I would instead use the term "dehydration."

Reviewer #3

(Remarks to the Author)

Overall the authors have made a number of changes to the manuscript to improve it, by editing the text and figures where appropriate, providing additional clarifications, as well as additional experiments. I have only a couple minor concerns remaining.

1) The text for Figure 1j does not seem to adequately explain the figure panel - there is no mention of G4r3a silencing in the text.

2) I appreciate the added text that now accompanies the model simulations, though I think an additional sentence needs to be added to explain the logic of why the authors chose the specific manipulations they did in representing the "sucrose

state" - decreased maximum intensity of LNa activity, and addition input to Handle-b neurons. I understand the goal of manipulating the activity state of different classes of inhibitory neurons, but there is no clear logic in the text to justify these specific manipulations.

3) In the new description of the supervised classifier, the statement "common actions may have very different features" is a bit confusing to me. Does this suggest that the same behavior looks different when in different feeding states? How, then, were the annotations performed? If they were blinded, for example, this would seem to cause some issue if the same behavior looks different.

Reviewer #4

(Remarks to the Author)

We thank the reviewers for their insightful comments on the manuscript. We have conducted additional experiments and analyses that address the major comments and weaknesses from the reviewers. The main work conducted consists of food quantification, behavioral and calcium imaging experiments to further characterize the internal state of the animals, calcium imaging and behavioral experiments and analyses that comprise neuropeptide and neuropeptide receptor manipulation to support the implication of sNPF and NPF signaling in feeding state-dependent modulation. We have additionally made changes in the manuscript text and have updated behavioral and transition probability analyses and statistical methods. We describe in detail all the additions and changes to the manuscript in the point by point response below.

REVIEWER COMMENTS

Reviewer #1 (Remarks to the Author):

The study by Tredern et al. explores how physiological states like hunger affect behavioral decisions, specifically focusing on the neural circuits involved in such state-dependent decisions using *Drosophila* larvae as the experimental model. The authors aim to understand how hunger and thirst influence non-feeding related behaviors such as threat avoidance, which involves choosing between different actions like "Hunching" (protective) and "Head Casting" (exploratory). The methods involved combining comprehensive circuit mapping using electron microscopy (EM) with targeted manipulations of identified neurons. Behavioral responses to an aversive air-puff stimulus were monitored using automated tracking and machine-learning-based classification to analyze the probability of different actions. The authors show that short-term food deprivation leads to increased locomotion and exploratory behaviors. Starved larvae and larvae fed only on sucrose exhibited more Head Casting and less Hunching compared to normally fed larvae. This behavioral shift was attributed to differential modulation of two reciprocally connected inhibitory neurons: one promoting Hunching and the other inhibiting it. Neuropeptides NPF and sNPF were identified as key modulators affecting the activity of these inhibitory neurons, thereby biasing the larvae's response towards more exploratory actions in a feeding state-dependent manner. Together, this work provides insights into how internal states influence behavior by altering neural circuit activity, demonstrating that hunger and thirst can modulate non-feeding related behaviors through specific neural mechanisms.

Overall, I find the subject of the manuscript to be interesting and relevant, and the paper is for the most part well written. The hypothesis that internal feeding states affect non-feeding behavioral programs via release of neuromodulatory hormones is elegant and attractive. However, the study lacks further experimentation and validation in certain parts of the study, which together prohibits me from recommending publication at this point. It should be further noted that although the work on larval behavior and the associated computational work seems very carefully planned and executed, this is not within my area of expertise and I have therefore not made detailed comments on these parts of the manuscript. I have outlined my concerns about the manuscript below.

Major comments:

1. In Fig. S1 the authors report that rehydration of larvae after feeding only on 20% sucrose restores their behavioral outputs to levels similar to that of controls. These data imply that the observed behavioral changes

reported are, at least partly, induced by osmotic disturbances rather than by an increased motivation to locate nutrients as the authors also later argue (L155-161). Indeed, 20% sucrose represents a major osmotic challenge to the animals. This raises two specific points: 1) it remains unresolved whether the increase in yeast consumption is due to an increase in protein-specific appetite (A2) or simply a result of the animals attempting to restore osmotic balance by consuming a more dilute medium (i.e. high water). 2) the authors should also make further attempts to separate osmotic and nutrition inputs on behavior by considering exposing animals to the same osmotic challenge using non-nutritive osmolytes (i.e. inorganic salts and/or sucralose to produce a medium with the same osmotic pressure as that of 20% sucrose) and test if they still observe the same reversion of behavioral outputs?

We thank the reviewer for the suggestion. Sucrose state is complex as the larvae fed on 20% sucrose would lack proteins, and water and in the same time have high glucose level. Dissecting how each of these factors contributes to the larval behavioral changes would indeed contribute to a comprehensive understanding of how nutritional state with high glucose levels and lack of other nutrients affects larval motivation and behavior. However, a complete dissection of the different types of inputs that could influence behavior is beyond the scope of the current study that focuses on dissecting the neural circuit mechanisms underlying sensorimotor decision modulation (during response to an aversive stimulus) as a function of internal states. We aimed to determine whether and if yes, where and how the feeding state affects processing of non-feeding related decisions. We specifically focused on reciprocally connected inhibitory neurons embedded in a connectivity motifs reciprocal inhibition of inhibition that was previously proposed by us and others to implement flexible competitive selection (Jovanic et al., 2016; Koyama and Pujala, 2018; Mysore and Kothari, 2020).

We have made this clearer by adding the following sentence in the introduction: “Reciprocal inhibition of inhibition has been proposed to be such a motif that could confer the circuits the property to be tuned to contextual/state information and thus implement flexible competitive selection (Jovanic et al., 2016; Koyama and Pujala, 2018; Mysore and Knudsen, 2012; Mysore and Kothari, 2020). However, the detailed mechanisms and implication in a case of state-dependent flexible selection haven’t been experimentally demonstrated”.

We thus undertook to explore the different internal states with the primary aim at determining that the relatively short food-deprivation protocols that we use (90 min) had an effect on non-feeding-related behavior and therefore are suited to be used in the investigation of the effects of how changes in feeding states affect sensorimotor decisions in response to aversive stimuli and specifically the neural circuit mechanisms at the level of the reciprocally connected inhibitory neurons in the characterized circuit.

Regarding the motivational drive of larvae fed on sucrose, the results of the locomotion analysis reveal that larvae fed on sucrose show increased exploration similarly to starved larvae (see Fig. 1a-d and Supplementary Fig. 1a-p). While this increased exploration could be motivated by a search for protein or water it still constitutes a motivation of searching for nutrients that could influence behavioral decisions in a similar way, as does the motivation of searching for food in starved larvae.

We performed several experiments to address the specific points raised:

Regarding the question of the increase in yeast consumption by sucrose-fed larvae, we have performed experiments where we tested yeast consumption in larvae fed on standard food and those fed on standard food with an addition of 20% sucrose. We found no difference in the amount of consumed yeast paste in larvae fed on normal food and larvae fed on food made with 20% sucrose. This suggests that the increase in intake of yeast paste in sucrose-only fed larvae (Fig. 1b) is also due to the lack of protein and not only water.

Reviewer figure 1. Consumption of yeast was similar between larvae fed on normal food (n=20 larvae) and those fed on normal food with an addition of 20% sucrose (n= 18 larvae) (two tailed t-test of equal variances, p-value=0.3023) (also included in Figure 1)

However, the experiments where sucrose-fed larvae were rehydrated for 15 min showed that rehydrated larvae reduced their exploration and behaved more similarly to normally fed larvae. These larvae were still lacking proteins which would suggest that the driving cause for the behavioral changes in sucrose-fed larvae could be the increased need for water.

Regarding the effects of other media of similar osmotic pressure we performed experiments where larvae were given sucralose of the same osmotic pressure as that of 20% sucrose solution for 90 min. The sucralose-fed larvae showed similar behavioral changes: they crawl faster, spend more time in runs and disperse more. This excludes that the high glucose hemolymph level is at the origin of changes in larval decisions. We have included these new experiments in Supplementary Fig. 1 (panels q-t and put the transition probabilities of sucrose fed and starved larvae in Supplementary Table 5).

The further dissection of these states and specifically investigating how osmotic changes affect larval behavioral decisions in more detail is an interesting question and will allow us to potentially determine putative neuronal and signaling pathways upstream of or parallel to NPF neurons that convey information about the change in the internal state to the ventral nerve cord. This is a topic of ongoing efforts in the lab that we have initiated since this paper and we hope to be able to elucidate in future studies

2. The method employed for quantifying food intake I am not familiar with, but I see some inherent limitations of this approach. Given that the authors only quantify the surface area of the gut occupied by the food (i.e. pixel number above background fluorescence) they don't collect information on pixel intensity, which would be

equally important of assessing the amount of food intake. Further, they employ an epifluorescent microscope to collect pictures in 2D this means that they won't capture areas of the gut that overlap with each other, which would lead to wrongful estimations. Finally, the approach is also very sensitive to changes in larval size (e.g. large larvae bigger guts) and given the signal is not normalized to total gut size this can create large variation. In general, all assays for estimating food intake have limitations, which means that you should always use multiple approaches to assess food intake. In my experience, only when several assays point in the same direction can you be sure of what is happening. I would strongly advise the authors to supplement their food intake experiments with another approach, e.g. the blue dye assay that is used successfully by many labs, to independently test food consumption.

We have used the fluorescence dye as we have initially tried alternative methods relying on measuring absorbance of larval homogenates. The values measured were very variable and the experiments required a large number of individuals and/ or high food dye concentration (which seems to impact food consumption, see below).

We have now also tried to use a different dye (Orange G) and measured absorbance. The experiments of quantifying food intake in fed and starved larvae confirmed the results of the image-based fluorescence quantification of ingested rhodamin dye.

Reviewer Figure 2. Starved animals increased their feeding on a standard food medium as compared to fed animals. (Two-tailed t-test of equal variances, n=4 samples of 5 larvae each in Fed, n=8 samples of 5 larvae each in Starved, **:p<0.01). p-value=0.0043439

For sucrose fed larvae the Orange G seemed aversive (as they ingested little medium with the dye and moved out of the dish). We tested this with a preference test where half of the agarose plate was coated with Orange G dye solution and monitored larvae locomotion in larvae fed on standard food and sucrose-fed larvae and observed that larvae fed on sucrose avoided the side of the dish coated with Orange G.

Review Figure 3. Larvae fed on 20% sucrose for 90 minutes avoid the area of agar plate with Orange G solution (n=39 for Fed, n=42 for Sucrose Fed at t=20s)

The quantification of rhodamin dye initially used in the paper indeed extracts the area of fluorescence above a certain threshold from the 2D image taken of the larva. To account for the amount of dye ingested we have now also quantified the intensity of fluorescence. The value that was used to quantify and compare food intake between groups is the product of the area and the mean intensity of fluorescence. Thus both the intensity and the area contribute to the quantification of food intake.

During data analysis, for each experimental day, the area is normalized by the average area of the control conditions, which are larvae fed on standard food on the same experimental day. The product of the normalized area and intensity is then computed. On each day, larvae of similar size were selected based on visual inspections and the normalization accounts at least for any experimental day variability that might exist in larva size.

We have clarified this in the methods section.

The results of food intake quantification give insight about larval physiological state and motivation. These are further supported by complementary behavioral experiments: locomotion analysis after starvation or sucrose treatment and then refeeding and rehydration respectively. These results are in line with the findings that the changes in motivation after 90 minutes of starvation or sucrose feeding are the results of lack of nutrients or water respectively (Fig. 1 and Supplementary Fig 1). In addition, the preference for sucrose assay confirms that larvae fed on high sucrose concentration avoid sucrose in line with their decreased intake of sucrose and high circulating glucose levels (Fig. 1i, j, l, m).

3. The authors propose that the NPF neurons are activated upon sugar-only feeding as well as on starvation (water only) relative to the normally fed larvae. This would imply that these neurons are able to integrated information, either directly or indirectly, on internal changes in sugar and water levels, as also shown for other neurons, albeit in the adult brain (1, 2). How might this be achieved mechanistically? Although the exact molecular players can be hard to identify, whether this is sensed autonomously or non-autonomously could be

interesting to determine and could be tested by silencing synaptic input using TTX. This might help to connect and integrate this work with future studies on central nutrient sensing in *Drosophila*.

Investigating how the NPF integrates changes in sugar and water levels would indeed shed insights on where and how are the changes in physiological states sensed by the nervous system. We believe, however, that such analysis is beyond the scope of this manuscript and would be better suited to future follow-up studies, as here we focus on the circuit mechanisms at the level of reciprocally connected inhibitory neurons that bias sensorimotor decisions. To address the specific point about NPF autonomously or non-autonomously sensing of nutrients we have performed the suggested experiment, that we describe below.

We typically perform calcium imaging experiments in different feeding conditions in intact larvae. We have specifically developed this preparation to image neuronal activity in different states to preserve the physiology of a state being investigated as a dissected preparation would disrupt the internal physiological environment of the animal. Since we could not apply TTX in intact larvae, we performed experiments in a semi-dissected (filet) preparation to expose the CNS. We recorded NPF in the different states but could not observe any significant difference in NPF activity in dissected preparations (Reviewer Figure 4). We therefore applied glucose directly to the dissected preparation to mimic changes in glucose levels before and after applying TTX and imaged NPF neurons. We found that applying glucose decreased NPF activity before and after TTX was applied suggesting that NPF could sense changes in glucose levels autonomously.

We have included these results in the Supplementary Fig. 8a and in the manuscript text (line 485-6)

Reviewer Figure 4. Monitoring GCaMP fluorescence in DM-NPF neurons reveals no significant differences in calcium levels among Fed, Sucrose-Fed, and Starved NPF-Gal4; UAS-GCaMP6s larvae, prepared as filets in an Artificial Hemolymph-Like (AHL) solution (ANOVA, $n = 10$ larvae, Turkey post-hoc test, Fed vs Sucrose Fed: p -value=0.899, Fed vs Starved: p -value=0.889, Sucrose Fed vs Starved: p -value=0.999)

Minor comments:

L122 "Crawling" is written with capital c. Is this on purpose?

Thank you for noticing. We corrected this

L151-152 If there is not a significant difference then you cannot claim that glucose levels are lower between fed and starved animals. Similar arguments are made throughout the manuscript (e.g. L255). Please adjust accordingly.

We have adjusted accordingly.

L419-423 I am curious that how the authors confirmed that the neurons were indeed NPF+ neurons? Was this solely done based similar morphology or was this done using the publicly available larval brain connectome?

Indeed we used the larval brain connectome. We have identified in the larval electron microscopy connectome NPF neurons as an upstream partner of handle-B neuron. We further matched these EM images to light microscopy images of the NPF-GAL4 line and also the Split GAL4 lines that we used in this study.

Figures.

Fig. 1: Mentioning figure panels in the text in a non-chronological order seems counter-intuitive and is likely to be subject to journal guidelines. Consider organizing data such that they are mentioned in chronological order, i.e. panel 1A before 1B etc.

Thank you for noticing. We have changed the panel referencing and made sure of the order of citing the figures in the manuscript.

Fig. 3. There is a black line near “hunch” in panel E that needs to be removed.

Thank you. We have corrected this

Fig1S: Here there is a pronounced difference in the average Max dispersion distance of the starved larvae from the 5h treatment (ca 26 m distance) and the 5h treatment + refeeding (ca 15 m distance). What is the reason for this relatively big difference between identical exposures? In general, using similar scale sizes for the same readouts across experiments would make comparisons much easier. Similarly, what might be the reasons for the markedly lower responses across all locomotor outputs in 18h starved vs 5h starved animals? Are these observations linked with nutrient depletion of the animals?

The 5h treatment and 5h treatment + refeeding experiments were performed at different times. Although the same genotype was used and the experiments were performed in controlled condition, it is not unusual to observe a variability between different days and between individuals. We thus mainly compare between conditions done at the same time, as we always perform all conditions within an experimental set in parallel.

Indeed the larvae that were starved for 18h seem to be slower and thus show lower dispersion and time spent in runs. This could be due to the lower energy in larvae after 18h starvation compared to larvae starved for 5h. However some of the metrics are lower in Fed control larvae in this set, so this could be due in part to variability between sets as discussed in the previous paragraph.

Regarding the comment on the scales: they were adjusted depending on the size of the plots, outliers etc. we have tried adjusted scales between experimental sets, but it made some comparison completely not visible as

it compressed too much the plots in some cases, we therefore chose to leave them adjusted to improve visibility for each plot, although we agree it is not ideal in terms of comparing plots between experimental sets. But given the aforementioned variability the more relevant comparisons are between the different conditions within the same experimental set, where the different feeding states are compared to the fed control within that experimental set..

Fig2S: in panel C the UA unit is not known to me. Is it meant a.u. as in arbitrary units?

Yes indeed, thanks for noticing. We corrected the axis legend to A.U.

1. Koyama T, Terhzaz S, Naseem MT, Nagy S, Rewitz K, Dow JAT, Davies SA, and Halberg KV. A nutrient-responsive hormonal circuit mediates an inter-tissue program regulating metabolic homeostasis in adult *Drosophila*. *Nature Communications* 12: 5178, 2021.
2. Jourjine N, Mullaney BC, Mann K, and Scott K. Coupled Sensing of Hunger and Thirst Signals Balances Sugar and Water Consumption. *Cell* 166: 855-866, 2016.

Reviewer #2 (Remarks to the Author):

This manuscript describes the neural basis of how the internal state influences the neural circuit controlling the responses to aversive mechanical stimuli in *Drosophila* larvae. The authors begin their investigation by examining how hunger affects the bias between two competing behaviors in response to air puff: hunching and bending. They reveal that the activities of groups of feedback or feedforward inhibitory neurons are influenced by the feeding state of the animal. Although the connectivity map of neurons underlying these behaviors is known, the authors discovered the nodes in this circuit that are subjected to modulation upon changes in the internal state. The authors discovered that a group of neuropeptide F (NPF) neurons is more active when the animal is hungry and modulates the activity of one of the feedback inhibitory neurons in the circuit. They also reveal that another neuropeptide, sNPF, inhibits the activity of these feedback inhibitory neurons, while it facilitates the activity of the feedforward neurons. In the era of connectomes, this study highlights the significance of neuromodulation in rendering neural circuits dynamic and adaptive to the context.

Overall, the study described in the manuscript is coherent and is a great follow-up to Jovanic and Schneider-Mizell et al. (2016) and Masson et al. (2020) studies. It is highly relevant to the field and will be of interest for the broad readership of *Nature Communications*.

Below are my notes and suggestions. I believe that a few additional experiments, some edits to the figures and the text, and some additional points of discussion would strengthen the work and make the reporting of the findings more concrete and clearer.

Major points:

1. In line 489 in the text, the authors state: “This increase in activity then biases the sensorimotor decisions towards less Hunching and more Head Casting.” This statement should be supplemented further with behavioral experiments by knocking-down NPFR in Hb neurons and measuring hunching/ bending.

We have now performed experiments where we knocked down NPFR in Hb neurons. Obtaining enough larvae for these experiments has been challenging as the combination of the split line that labels Hb (SS00888) and the line for the RNAi-knockdown gives very few progeny as they reproduce poorly. Additionally, the Split line is balanced both on the second and third chromosome (or is not viable) so typically only a quarter of the progeny can be tested. While this is less of an issue for calcium imaging experiments where about 10 larvae per condition are tested, it becomes a difficulty for behavioral experiments where we typically need hundreds of larvae per condition to allow for proper analysis and statistical comparison. Therefore, we did not fine-tune the classifier as the low number of training and testing examples would lead to over- or under-fitting depending on the testing-to-training ratio. Furthermore, we noticed the strong differences in locomotion parameters between the progeny genotype and the genotype of the original classifiers that would prevent proper transfer learning or fine tuning and would require full retraining.

Therefore, we used another statistical testing procedure that we recently developed (Blanc et al., 2024). The method relies first on an unsupervised learning procedure that does not provide discrete behavior but rather a low-dimensional continuous embedding of the larva motion. This embedding is not dependent on the larva genotype. Statistical testing is then performed by comparing the distribution of larva dynamics in that latent space using the kernel-based Maximum Mean Discrepancy (MMD) approach. We successfully implemented this approach to compare differences between different feeding conditions in control larvae and larvae where we knockdown NPFR in Hb.

We found that, as expected, there were differences in control larvae (SS00888>w1118, RNAi control) between fed and sucrose-fed and starved conditions. There were, however, not statistically significant differences between different conditions in larvae where NPFR1 was knocked down in Hb neurons (SS00888>NPFR-RNAi). Although we cannot specifically assign this phenotype to the changes in Hunch and Head casting, these results point to the role of NPFR in Handle-b in modulated air puff responses in a state dependent manner. In addition, the analysis was performed early upon stimulus onset, when the majority of actions are Hunching or Head Casting (Jovanic et al., 2016; Masson et al., 2020)

We have added these results in Supplementary figure 9 and incorporated them in the results section in the manuscript. We have added a brief description of the unsupervised method in the methods section.

2. Figure 6F: A more proper no RNAi control for this experiment would be Gal4> UAS-GCamp; empty RNAi vector (a non-targeting RNAi, i.e., the valium vector for the TRIP collection).

We have performed these experiments with the empty RNAi control in normally fed larvae and sucrose-fed larvae and have obtained similar results. In the empty RNAi control, we observe an increase in Hb responses upon feeding on 20% sucrose as we do in the no RNAi control. Upon NPFR knockdown there are no significant differences between larvae in different feeding conditions (Reviewer Figure 5)

Reviewer figure 5. Calcium responses in Handle-b upon NPFR knockdown compared to an empty-RNAi control (Handle-b>w1118, RNAi control). For all groups, n = 10 larvae, 1 recording per larva. In control larvae, sucrose feeding leads to an increase in Handle-b activity compared to standard food feeding (Mann-Whitney p = 0.03836). Knockdown of NPFR in Handle-b abolishes its increase in activity upon sucrose feeding (Mann-Whitney p = 0.36317).

3. Inferring from the sNPF-R knock-down experiments in Figure 7, sNPF plays a satiety signal. This notion should be tested with sNPF or sNPF-R knockdown experiments in starved animals to see whether they exhibit fed-like behaviors (i.e., hunch and bends).

We have knocked down sNPF-R using a panneuronal driver. We found that while in the control the Hunching is decreased and Head Casting increases upon starvation and sucrose only feeding in control feeding, the larvae where the sNPF-R was knocked down overall respond less to the air puff (in all states). The Hunching was more strongly decreased in fed larvae and thus increased in larvae fed on sucrose and starved larvae compared to the fed, suggesting that sNPF signaling is required for feeding state dependent modulation of the sensorimotor response to the air puff.

However since the sNPF-R is expressed widely in the nervous system, these results cannot be specifically linked to the modulation of the decision circuit we have investigated in this study.

Reviewer figure 6. Comparison of feeding state influence on air puff response in control larvae and larvae with brain-wide sNPFR knockdown. (Control fed n = 260 ; Control sucrose fed n = 213 ; Control starved n = 204 ; GMR_57C10 fed n = 356 ; GMR_57C10 sucrose fed n = 298 ; GMR_57C10 starved n = 314). sNPFR signaling is required for the decrease in hunching response upon sucrose feeding or starvation. (Chi-square tests for control fed vs control sucrose fed: hunch p = 0.011, static bend p = 0.774, head-cast p = 0.050 ; Chi-square tests for control fed vs control starved: hunch p = 0.030, static bend p = 0.004, head-cast p = 0.001 ; Chi-square tests for GMR_57C10 fed vs GMR_57C10 sucrose fed: hunch p = 0.009, static bend p = 0.094, head-cast p = 0.001 ; Chi-square tests for GMR_57C10 fed vs GMR_57C10 starved hunch p = 0.008, static bend p = 0.0489, head-cast p = 0.001)

We have therefore also knocked down sNPFR in G2 and Hb neurons and monitored decisions. Based on Calcium-imaging results in fed larvae, where sNPFR knockdown mimicked the effect of sucrose feeding and starvation, we postulated that sNPFR is downregulated in these conditions (Figure 7). As expected under this hypothesis, knocking down sNPFR in G2 and Hb in those states also had little effect on larval behavioral response probabilities to air puff, as it did on calcium-imaging responses of the two neurons (Supplementary Figures 10 and 11) . We have included these results in Supplementary figures 10 and 11.

4. The authors should present the individual data point throughout instead of simply using bar graphs.

We have now presented the individual data points wherever possible and specifically in the calcium-imaging plots and in feeding assay plots throughout the manuscript

5. In Figure 1 panel B1, the authors show that glycemia is increased in sucrose-fed animals and it does not change in starved animals compared to the control-fed animals. Yet, in the text, the authors conclude that behavioral changes they observe is 'due to lack of nutrients and lower energy levels' (lines 163-165). This interpretation should be revised. On that note, how do the authors explain their findings that sucrose feeding

and starving the animals produce the same behavioral outcome although they affect glycemia differently? This should be addressed in the discussion.

We have updated this statement by removing the lower energy levels claim. As stated (lines 174-176) the lack of nutrients (including water) results in an increased motivational drive that translates in increased exploration. This increased motivation and exploration is similar in starved and sucrose fed (and sucralose fed) larvae as quantified in Figures 1a-d and Supplementary Fig. 1a-p. These changes in the internal motivational drive to search (for lacking nutrients) would then result in changes in responses to aversive stimuli: increased active actions (Head Casting) and decreased static actions (Hunching).

We have clarified this in discussion (lines 616-644).

6. In Figure 3 and Extended Figure 3, Basin 2 neurons show differential activity in sucrose-fed and starved states: there is an increase in activity in sucrose-fed animals and decrease in starved (although not significant). How do the authors explain that these two states lead to the same behavior (i.e., increased bending in Figure 1 E1)? This should be discussed in discussion.

Indeed the Basin-2 activity at lower intensities of stimulation is not increased in the starved state (as it is for the sucrose-fed). Based on the model and the circuit, Basin-2 inhibits the Hunch and increases the Bend. Upon starvation, we also do not always see a significant decrease in Hunching (although a trend can be consistently seen), but we always see an increase in Bending. An increase in Basin-2 upon starvation can however be seen at higher intensities of stimulation. The decrease in LnA responses in the starved condition is also less pronounced at lower intensities of stimulation compared to the sucrose-fed condition. So it could be that the effects of starvation on the circuit are less strong and especially so at lower intensities of stimulation, leading to a milder modulation of neuronal activity and behavior (as we observed). This would then result in a weaker Hunching phenotype compared to the sucrose-fed larvae. The probability of failed response Basin-2 starts to be affected at medium stimulus intensities. The effects of sucrose feeding and starvation were tested and observed at high air puff intensities which more closely matches the high stimulus condition in imaging where an increase in Basin-2 can start to be seen.

The stronger phenotype upon sucrose feeding than starvation could be due to a stronger drive to search for water upon 90 min sucrose feeding than starvation (as thirst may be a stronger motivator than hunger, see also below).

We have added a paragraph on the differences between the effects of sucrose feeding and starvation in the discussion (lines 625-638).

7. Comparing Figures 4 A-B to Extended Figures 4 A-B. How do the authors explain that the differences in calcium activity are more prominent in sucrose-fed animals compared to starved animals? Given that both Griddle-2 and Hande-B neurons are modulated by the feeding state of the animal, the expectation would be that total starvation would have greater effect on the activity of these neurons compared to the sucrose only-feeding. This should be discussed in the discussion.

We were also initially surprised by this observation. Since we have found that rehydrating larvae for 15 min was sufficient to restore the control-like phenotype, it would appear that these larvae in addition to lacking proteins are also lacking water (Fig. 1 e-j) and Supplementary Fig.1 m-p). Our interpretation thus is that the need for water represents a stronger driver than the lack of food upon 90 min of treatment as lacking water becomes critical after a shorter time as longer treatments on 20% sucrose result in larvae dying. Alternatively, the lack of water, protein and increased glycemia in sucrose-fed larvae could all add up to have a stronger immediate effect on neuronal activity and behavior compared to starved larvae.

We have added this point in the discussion (lines 658-632).

Minor points:

1. To increase clarity and make the paper more fluent and easier to read, the figures should be put in the order of their mentions in the text.

Thank you for noticing this. We have made sure that the figures are cited in order.

2. Extended figures 1B-D median lines for controls are not visible.

Thank you. We have corrected this

3. Figure 1E- the error bars are missing.

We have added confidence intervals

4. Lines 330-332: "The increase in Griddle-2 responses upon Handle-b inactivation in larvae fed on sucrose did not reach the levels of Griddle-2 responses upon inactivation in larvae fed on standard food (Fig.4C2-C3, Extended Data Fig. 4E)." This is not clear from the figures mentioned. Lack of significance should be indicated in Fig. 4C3.

We have added the annotation of the lack of significance in Figures 4n and t

5. Line 375: "As expected, silencing Griddle-2 abolished the difference in Basin-2 responses in the two States." It is unclear to which figure this text refers. This should be specified. Assuming that it is Figure 5C, the lack of significance should be pointed out.

Thank you for noticing, we have made these changes (added figure number in the text and added the annotation for the lack of significance on the plot).

6. Extended Figure 3C: Basin 1 activity-sucrose-fed trace should be red.

We have corrected this.

7. Figure 6A the EM image: what do the lines in different colors show? This should be addressed in the figure legends.

We have added the explanation for the annotations of the arrows of different colors in the figure legend 6A (now Fig 6c).

Reviewer #3 (Remarks to the Author):

Summary

In Tredern et al., the authors address important questions in neuroscience: how does the internal state of an animal modify behavior and, on a mechanistic level, how are neural circuits flexibly modified by the state to support these changes. To address this, they investigate the influence of hunger on the behavior and neurophysiology of *Drosophila* larvae, focusing specifically on modulation within a circumscribed neural circuit involved in escape decisions upon mechanosensory stimulation. This circuit was previously identified by members of the same group as the current paper, and the current paper extends this previous study in an interesting direction. While I find the overarching theme of the work exciting and the general strategy and organization of the manuscript sensible, I found multiple aspects of the results and/or the authors' interpretations difficult to follow, and generally found the manuscript lacking concision. I clarify these and other issues below.

Major concerns:

1) The identification of "head casting" as an exploratory action seems an important detail of the behavioral description, but it seems that the distinction between this behavior and bending is not so clear throughout. At times, it seems the two terms are used interchangeably, which is confusing (e.g. line 280 says "bend" whereas line 296 says "head casting"). This becomes more critical when describing the circuit, where the focus becomes on the influence on basin-2 activity, which promotes bending. Is there any evidence that basin-2 activity is also related to "head casting"? If not, it's not so clear to me why the authors focus on "head casting" behaviorally. Related to this, why do the authors not also include "static bends" in their behavioral transition plots (e.g. in Figure 1E, right side)?

The change in terms for behaviors Bending and Head Casting is related to methodological differences in classifying behavior with the machine learning based classification (Lehman et al., 2025; Masson et al., 2020, this paper), compared to the original paper where the circuit is described for the first time (Jovanic et al, 2016). In this study, the behavioral definition (due to the difference in methods) of Bending overlap by large with Head Casting, rather than Static bend events, whereas the machine- learning based classification detected different types of Bending events (Masson et al, 2020). We have therefore in recent work separated the behavior in Head Cast and Static Bend (Lehman et al, 2025). To match the circuit motifs to the behavioral categories from the machine -learning classification and thus the Bend (from Jovanic et al, 2016) and Head

Cast can be used interchangeably. We have confirmed that Basin-2 is required for Head-Casting and not Static Bending by applying the machine learning algorithms to the Basin-2 silencing data (Reviewer Figure 7, and Supplementary Figure 2 k-m in the manuscript).

The Static Bend from our observation is a type of static response, and not an exploratory behavior as Head cast. It never occurs in absence of stimulation. The changes in Static-Bending, similar to Stopping, seem to be more correlated with Hunching than Head Casting. However we do not have a good understanding of the neuronal control of Static bending and therefore we do not include it in the transition plots for clarity (adding a fourth action would make the transition plots less clear and we believe would not add to our description of circuit mechanisms. All transitions are included in Supplementary tables. We have also clarified the use of Head Cast instead of Bend in the manuscript (lines 259-265)

Reviewer figure 7. Analysis of the Basin-2 silencing experiments. In response to air-puff larvae with silenced Basin-2 neurons Hunch more and Head-Cast less, confirming that Basin-2 is required for Head-Casting and not Static bending, the new defined actions based on the previously defined action Bending

2) It is not clear why the authors present multiple types of hunger states in Figure 1 (sucrose-fed and starved) only to later primarily focus on the sucrose-fed state. As a reader, it is unclear why one state was favored over the other in later figures. The wording describing the Figure 1 results, for example, lines 139-148, is not straightforward, and it is a bit frustrating if the distinction between the two states is not clearly useful for later sections. In figure 2, for example, the same experiments were performed in both hunger states, but the starved data is buried in a supplement which makes it difficult to directly compare. By the time we get to figure 3, the starved state is no longer addressed. Are the differences observed between the two states relevant for the major findings of the paper? If so, how?

The main findings of the circuit mechanisms (and neuropeptidergic pathways involved) underlying the effects of these two feeding states on sensorimotor decisions seem to be shared between the two states. As mentioned above we believe that this is due to a common underlying motivational state: increased exploration behavior due to increased search drive for lacking nutrients (likely water and possibly proteins in the case of sucrose-fed larvae and food for starved larvae (see added paragraph in discussion). So while there are undoubtedly differences in the way the different states are sensed and decoded in the central brain the downstream mechanisms that regulate locomotion and behavioral responses to aversive stimuli seem to converge onto the same circuitry and share similar mechanisms that then influence similarly behavioral output.

We have performed most of the experiments in three states (normally fed, sucrose fed and starved, except for the functional connectivity experiments(i.e in Figure 4k-n,q-t, Figure 5, Supplementary fig. 5 g, h, Supplementary fig. 6) where the goal was to determine whether the state-dependent changes that we observe in the different neurons of the circuits are due to the direct modulation of each neuron or due to the modulation of their presynaptic partners. We reasoned that given that we know how the activity changes in each neuron for the different states, that the functional interactions we observe in the sucrose state would be similar in the starved state.

We chose to focus on the sucrose-fed state in these experiments rather than the starved state as the effect of the sucrose-fed state is somewhat stronger, possibly due to the dehydration being more critical than starvation at this duration of treatment.

We have added text in discussion of the manuscript that clarifies these points (614-641).

3) The modelling data presented in Figures 3 and 4 is poorly explained, both in the results section as well as in the figure legends. As a reader, I was not able to grasp their significance, and many of the plots contain multiple features that are simply not discussed anywhere. For example, in Figure 3C2, D2, E2, what is w_{iLN_a} ? What are the different lines? What is interesting or significant about the differences between the dotted (sucrose) versus the solid (fed) lines? In 3C3, D3, what is “affinity of input signal to iLN_a ”? This is not explained anywhere. Lines 290-292 mention “the sucrose state was modeled in one version of the model as...” but how does this translate to what is being shown in for example Figure 3C3, D3? My guess was at the lower r_{max} in figure 3C3 and the higher input current in 3D3, but I just wasn't sure I was reading this plots properly. I have the same issue with the model data in Figure 4C. These data are simply not explained in the text sufficiently for a reader to understand. Moreover, the specific models explored in Figure 3 seem rather contrived, basically matching the physiological results that follow in the subsequent figures. Some of the findings of the modeling also seem a bit trivial given the basic circuit (e.g. the silencing of Handle-b neurons described in lines 365-372). Are there alternative models that should be considered? As is, the models feel like throw-ins that do not add clarity or insight to the story, but rather force the reader to figure out how to interpret them on their own.

Thank you for pointing out a weakness of the manuscript. We expanded the section “Reciprocally connected Inhibitory interneurons in the decision circuit are differentially modulated by the feeding state”, by lengthening the second paragraph and adding two paragraphs below it, to accurately describe the meaning of the figures and better explain their relevance to the overall study. We also expanded the legends to Figures 3 and 4 to include more information. Most of the details provided are repeats from the “materials and methods” sections, but we agree that including them in the main text and legends facilitates comprehension. We agree that the models proposed have severe limitations, and, in fact, acknowledged in the manuscript that they don't allow us to falsify either hypothesis. We chose to use these models as they are simple extensions of the model proposed in the paper that first studied the circuit. This original model is thus a reference point for our discussion. Additionally, we think that a simple, qualitative model is useful insofar as it validates the possibility for the proposed mechanisms to explain some of the experimental data, particularly since more detailed models would be hard to constrain. While more detailed modeling frameworks could be used in the future,

along with for example electrophysiology data to properly constrain them, they are outside the scope of this particular piece of work and we believe are better suited for future studies.

The expanding explanation of the model in the section “Reciprocally connected Inhibitory interneurons in the decision circuit are differentially modulated by the feeding state” corresponds to lines 290-351

4) There are multiple aspects of the results where the authors’ interpretation is unclear and/or hand-wavy. These interpretations sound speculative and do not add to the paper, but rather are clumsy to read through and distracting. For example, lines 267-271, 332-336, 347-350, 389-399, 408-411, 441-449, (sounds list-like), 467-471 (hard to follow/not helpful). Related to lines 389-399, the result that silencing Griddle-2 decreased Basin-2 responses when sugar fed struck me as really odd – the authors make note of this (lines 378-379), but I simply could not follow their subsequent interpretation.

Thank you for pointing this out. We have rephrased or removed these in the revised manuscript. For example, we have rephrased 332-336, 441-449, 467-471 and we have removed statement 267-271, 347-350, 389-399 and 408-411 and placed similar explanations in discussion where appropriate.

The interpretation of the effect of silencing of Griddle-2 on Basin-2 responses relies on knowledge of previous description of the circuit (Jovanic et al., 2016). This work has showed that Griddle-2 disinhibits Basin-1 through feedforward inhibition and that the Basin-1 could disinhibit Basin-2, through later disinhibition (via Handle-a), to trigger a transition from the Basin-1 only state to the Co-active state (that would correspond to the Hunch to Bend sequence). Thus Basin-2 are activated both by direct mechanosensory input and also can be activated upon Basin-1 activation. The occurrence of the coactive -state will depend on both these inputs.

To explain the reduction in Basin-2 responses upon Griddle-2 silencing we speculated that since silencing Griddle-2 will also disrupt the disinhibition of Basin-1 and therefore there will be less instances of Basin-1 to Basin-2 disinhibition, which essentially would only leave the Basin-2 activation by direct mechanosensory input. At the same time, inactivating Griddle-2 increases Handle-b responses and thus amplifies Basin-2 responses more once Basin-2 responses are triggered. In sucrose fed larvae since Griddle-2 activity is already low (and Griddle-2 is not inhibiting Basin-2 strongly) other inhibitory pathways may be favored and result in lower Basin-2 responses.

We acknowledge that these explanations remain speculative and may be distracting. We have thus simplified the explanation for the differences in results for sucrose fed and fed larvae in the manuscript result section.

5) It seems that the authors should instead do a Handle-b activation experiment instead of silencing to more explicitly test the hypothesis presented in lines 456-460 (related to figures 6B, D)? As is, it’s not clear why the Handle-b silencing experiments (lines 451-460) were not done in a sucrose-fed state, where Handle-b activity was shown to be high.

We have performed both Handle-b inactivation and activation experiments (while imaging in Basin-2 upon these manipulation). Those results are shown in Figure 5 and Supplementary Figure 6. In Figure 6D, we have silenced the NPF neurons and imaged activity in Handle-b. The goal of these experiments was to functionally

test the connection between NPF and Handle-b neurons that we identified in EM and based on immunohistochemistry and make sure that it is functional. We have initially tried performing NPF optogenetic activation experiments (with Chrimson) and imaging in Handle-b. However, the different light stimulation protocols we used didn't result in the activation of NPF neurons. We therefore focused on inactivation experiments to show that these neurons and the connections we identified structurally were functional. We use KIR to hyperpolarize DM-NPF to disrupt both synaptic and neuropeptidergic communications between NPF and Handle-B neurons.

Since the goal was to test whether the connection was functional we only showed these experiments for the fed state (the reference state). Also, since KIR hyperpolarizes DM-NPF neurons constitutively, our reasoning was that the effects of the amplitudes of effects of this manipulation in different states may not be best assessed by these experiments due to possible adaptation mechanisms.

We now added the results of these same experiments performed in starved and sucrose-fed when NPF activity and Handle-B is higher (In Supplementary figure 8). In starved larvae, we see a stronger decrease in Handle-b neurons upon silencing. In sucrose-fed larvae the decrease was smaller mainly to what appears to be a more moderate increase in Handle-B response in these larvae (compared to fed) in this set of experiments. Nonetheless experiments in all three conditions show that silencing NPF neurons decreases Handle-b responses and that the NPF Handle-b connection is functional.

6) The NPF-R staining shown in Figure 6E is not convincing. Can this be somehow quantified? Alternatively, could an intersectional strategy be used that includes LexAop-flp and UAS>stop>GFP?

The immunohistostaining experiment relied on staining against GFP (to label handle-N neurons as the Handle-B specific driver will drive GFP expression in these neurons and DsRED to label NPFR positive neurons as the LexA was inserted into an intronic region of the NPFR gene using a T2A (Lee et al., 2018.) and will drive transcription in neurons where the NPFR1 gene is normally transcribed. Colocalisation of the GFP and DsRed confirms that the Handle-b expresses NPFR1. We have made new images that show more clearly the colocalisation of the two antibodies in the same neurons. We have made stacks with fewer and more tightly spaced slices that cover only the volume of the neurons of interest (Figure 6l-p). In addition, we include here also the images made with only 2 successive slices covering 0.4 um of thickness that clearly show that the Handle-B neuron is positive for NPF-R (Reviewer figure 8).

Reviewer Figure 8. Here each image is 0.4 μm thick (stack of 2 successive slices) taken in the middle of each neuron N1 to N4. It shows the NPF-R positive staining of each SS00888>GFP neuron present in the main image of the article. Scale bar: 5 μm .

Another point supporting the Handle-b expresses NPFR is that the DNA sequence of the genetic drivers that label Handle-b neurons in different GAL4 and LexA (i.e. R60E02 that we show in this study, among others) is associated with the NPFR gene (FBgn0037408) based on publicly available databases (FlyBase, Bloomington stock center, for example stock number: 39250).

Other concerns:

1) The figures are not organized in line with the text, and this makes things difficult to follow. For example, in Figure 1, panel D is discussed before panel A in the text. Similarly, in Figure 5, panel B is discussed before panel A. This may occur elsewhere, but these two examples caught my attention.

Thank you for pointing this out. We have corrected this

2) Many of the panels also have multiple sub-panels that do not have unique identifiers. This makes it difficult to read the text and know which specific sub-panel is being referenced.

Thank you for pointing this out. We have annotated each panel with a unique identifier.

3) The axes on some plots are too zoomed out to discern the data. In Figure 1D for example, the axes make it difficult to assess the different groups. This also makes it hard to discern the RNAi results described in Figure 6C.

For the axes, we tried to have the optimal scaling given the space constraints that would allow to include outliers and to allow space to significance. All the data in the plots is available as source data in Supplementary table 3 that allows the comparison of actual values. The p-values are shown in Supplementary table 2

4) Why was “scaled speed” selected as the exemplar phenotype for the RNAi results in Figure 6C? Why not one of the others?

Given the space constraints, we chose to only show one plot. We have selected only one metric and we put the remaining metrics in the supplementary data. We chose the scaled speed as it appeared to be the metric that can be most easily interpreted as a standalone metric. Tortuosity and time spent in runs are both reflecting the Head Casting and Crawling ratios so it would make sense to keep them together. The Maximum dispersion is a result of changes in speed and the time spent Head Casting and Crawling. The other locomotion metrics are included in the Supplementary figure 8

5) Comparing the inactivation data between fed and sucrose fed seems the major point of figure 3C3, but statistics were not performed between these two groups (the magenta outlined plots).

Thank you for noticing this. We have added this comparison in Figure 4n (as we believe the comment referred to this panel and figure, not Figure 3).

6) I find the control experiment described in lines 221-224 unorthodox. Is there a reference for this approach?

We have found previous use of that approach. In their 2006 paper, Goentoro et al. showed that protein quantification via GFP fluorescence closely reflects transcript expression quantified via qRT-PCR (Goentoro et al., 2006). In this control experiment, because we were able to image several animals from each feeding condition during a single imaging session, we were able to keep the imaging conditions identical across animals so that differences in fluorescence levels would only come from differences in GFP expression. Since we could not see any difference in fluorescence levels across feeding conditions, we conclude that Gal4-UAS controlled transcription in these neurons are not affected by our feeding manipulation.

7) I am also not so familiar of making comparisons of GCaMP raw fluorescence across animals, as was done in Figure 6B (lines 430-435). Is there a reference for this approach?

The main obstacles in using raw GCaMP fluorescence to compare the effect of a manipulation across animals are variabilities at the level of fluorescent protein expression, imaging parameters and animal preparation. Since we are comparing same GAL4 drivers driving GCaMP in different feeding conditions and since we were able to show that our feeding manipulation does not influence Gal4 expression levels, there should be no difference in expression GCaMP level between different larvae. Moreover, we took care to record animals from all feeding conditions in the same imaging session, and kept all imaging parameters identical. Finally, we made sure to record the same number of animals for each feeding condition, each experimental day, as to make sure to not introduce any day to day variability in the comparison. We thus believe that the differences in raw fluorescence we found in this experiment reflect differences in the activity of the NPF neuron depending on the feeding state.

An example of previous study using the raw GCaMP as a measure of activity is Streit et al., 2016, albeit comparing across left right side and segments

8) When introducing the behavioral classification analysis, it could be helpful to mention whether these are supervised or unsupervised algorithms (lines 188-196). The methods describing the “Behavioral classification analysis” is not very concise and difficult to read. The part about “static bends” versus “head casting” should perhaps be detailed in a supplementary figure as the text alone is not clear (lines 857-861).

The behavioral classification method that we used is a supervised machine-learning-based method we have developed in a previous study (Masson et al., 2020) and have adapted it here.

We have specified in the text that the method is supervised. We have edited the behavioral classification method section to make it clearer. We specifically clarified the Head Cast and Static Bend separation sections (lines 892-896, 902-914) and added text about this in the manuscript (lines 200-214, 259-265). The definition of actions Head-Cast and Static Bend is detailed in a recently published study (Lehman et al., 2025) that we now refer to here.

In the methods, we also detail the fine-tuning procedures used in this study, including the new features associated with larva behavior, to adapt the classifiers to genotypes and the different internal states that exhibit phenotypes statistically different from the original training dataset.

9) Is it possible for the violin plots (e.g. Figure 2B) to include indications of the individual data points? I have seen this done with horizontal lines. This could interpret these plots, and the number of animals, easier to decipher.

We have added individual data points for violin plots throughout the manuscript

10) When introducing the mechanical stimulus (starting on line 258), it's not clear that a range was used as the

results only mention 5V, but the figure shows a range. Also, the text in line 260 seems it should reference figure 3B as well.

We have specified the stimulus intensities used in this section and also added the reference to Figure 3f (previously Figure 3 B) were relevant

11) The results paragraph on lines 198-204 should mention that a 5s stimulus was used.

The stimulus used in the experiments was 30 seconds as specified in the method section. The 5s refer to time window over which the behavioral probabilities were computed. We analyze only the first five seconds to cover the initial response where hunching occurs. The time windows are specified in the legends and in the method section of the manuscript.

12) What is the significance of the statement on lines 203-204?

Extending the food treatment still results in the same phenotypes, suggesting that these changes in behavior are not specific to 1h30 duration of diet perturbation but also to longer duration to several hours. We have added a clarification in the text

13) "Sucrose fed" is sometimes referred to as "protein-deprived" (e.g. in Figure 1C, Figure 5B) and this was quite confusing. The authors need to be consistent with this to avoid confusion.

Thank you for noticing this. We have corrected this.

14) Typo in Figure 1 legend: first five seconds "uoin" stimulus onset

We have corrected this. Thank you

15) Figure 2C doesn't mention which stimulus intensity is being shown (neither in the figure nor in the legend).

Thank you for noticing this, we have added this in the legend and expanded the method section to specify the detail about optogenetic stimulation.

Reviewer #4 (Remarks to the Author):

References

- Goentoro LA, Yakoby N, Goodhouse J, Schüpbach T, Shvartsman SY. 2006. Quantitative analysis of the GAL4/UAS system in *Drosophila* oogenesis. *Genes N Y N 2000* **44**:66–74. doi:10.1002/gene.20184
- Jovanic T, Schneider-Mizell CM, Shao M, Masson J-B, Denisov G, Fetter RD, Mensh BD, Truman JW, Cardona A, Zlatic M. 2016. Competitive Disinhibition Mediates Behavioral Choice and Sequences in *Drosophila*. *Cell* **167**:858-870.e19. doi:10.1016/j.cell.2016.09.009
- Koyama M, Pujala A. 2018. Mutual inhibition of lateral inhibition: a network motif for an elementary computation in the brain. *Curr Opin Neurobiol* **49**:69–74. doi:10.1016/j.conb.2017.12.019
- Lee P-T, Zirin J, Kanca O, Lin W-W, Schulze KL, Li-Kroeger D, Tao R, Devereaux C, Hu Y, Chung V, Fang Y, He Y, Pan H, Ge M, Zuo Z, Housden BE, Mohr SE, Yamamoto S, Levis RW, Spradling AC, Perrimon N, Bellen HJ. n.d. A gene-specific T2A-GAL4 library for *Drosophila*. *eLife* **7**:e35574. doi:10.7554/eLife.35574
- Lehman M, Barré C, Hasan MA, Flament B, Autran S, Dhiman N, Soba P, Masson J-B, Jovanic T. 2025. Neural circuits underlying context-dependent competition between defensive actions in *Drosophila* larvae. *Nat Commun* **16**:1120. doi:10.1038/s41467-025-56185-2
- Masson J-B, Laurent F, Cardona A, Barre C, Skatchkovsky N, Zlatic M, Jovanic T. 2020. Identifying neural substrates of competitive interactions and sequence transitions during mechanosensory responses in *Drosophila*. *Plos Genet* **16**:e1008589. doi:10.1371/journal.pgen.1008589
- Mysore SP, Knudsen EI. 2012. Reciprocal Inhibition of Inhibition: A Circuit Motif for Flexible Categorization in Stimulus Selection. *Neuron* **73**:193–205. doi:10.1016/j.neuron.2011.10.037
- Mysore SP, Kothari NB. 2020. Mechanisms of competitive selection: A canonical neural circuit framework. *eLife* **9**. doi:10.7554/eLife.51473
- Streit AK, Fan YN, Masullo L, Baines RA. 2016. Calcium Imaging of Neuronal Activity in *Drosophila* Can Identify Anticonvulsive Compounds. *PLoS ONE* **11**:e0148461. doi:10.1371/journal.pone.0148461

RESPONSE TO REVIEWERS' COMMENTS

We would like to thank the reviewers for spending time in thoroughly evaluating the revised manuscript and were satisfied with the changes that were made to address the concerns that were raised. The final version has taken into consideration the remaining comments from reviewer 2 and 3. We detail the point by point response below

Reviewer #1 (Remarks to the Author):

The authors have, within reason, addressed all comments and suggestions and I therefore have no further points to raise with this manuscript.

Reviewer #2 (Remarks to the Author):

I am overall satisfied with the authors' revisions and would recommend accepting the manuscript for publication. That said, I have a few very minor suggestions for text edits that in my opinion would enhance the argument and clarity.

1. The new text added to the second paragraph of the Results section needs a better transition. As it is standing now it is abrupt and reads a bit of a non-sequitur. A transition sentence that explains the rationale (i.e., checking the possibility of the effect of stress due to the osmolarity of 20% sucrose).

Thank you for the suggestion. We have added a transition sentence:

“To verify if larvae experience osmotic stress upon sucrose feeding , water consumption was quantified in the different states. “

2. The text discussing Supplementary Fig. 9 on page 11 is cursory. The data supports the arguments, and a bit more detailed description would help.

We have expanded the text that now reads:

“This outcome was reflected in the behavioral responses to an air puff, where upon NPFR1 knockdown in Handb neurons, no differences in behavioral responses could be observed between the three different states, i.e., fed, sucrose only and starved, further suggesting that NPF signaling is required for the state-dependent modulation of these neurons (Supplementary Fig. 9).”

3. In the new text in the Discussion section (page 13) the term “lack of water” is awkward. I would instead use the term “dehydration.”

Thank you for the suggestion. We have modified this and it now reads:

“The stronger effect of sucrose only feeding on neuronal activity and behavior could be due to dehydration being a more critical state than hunger (after 90 min of treatments). “

Reviewer #3 (Remarks to the Author):

Overall the authors have made a number of changes to the manuscript to improve it, by editing the text and figures where appropriate, providing additional clarifications, as well as additional experiments. I have only a couple minor concerns remaining.

1) The text for Figure 1j does not seem to adequately explain the figure panel - there is no mention of G4r3a silencing in the text.

Thank you for noticing. We have corrected this. We have added the GR43a silencing in the text.

2) I appreciate the added text that now accompanies the model simulations, though I think an additional sentence needs to be added to explain the logic of why the authors chose the specific manipulations they did in representing the "sucrose state" - decreased maximum intensity of LNa activity, and addition input to Handle-b neurons. I understand the goal of manipulating the activity state of different classes of inhibitory neurons, but there is no clear logic in the text to justify these specific manipulations.

We have clarified the goal of manipulating the inhibitory neurons to represent the sucrose state by adding the following text. :

“Based on the calcium imaging results (Figure 3-a-g, Supplementary Fig. 4g-p) we manipulated the activity state of the reciprocally connected feedforward and feedback inhibitory neurons in the circuit: LNa and Handle-b (Hb) neurons to explore whether the modulation of one of the neurons was sufficient to bias the state-dependent changes in neuronal activity and behaviors or the modulation of both neurons was required. In one version of the model, the sucrose state was modeled as the decreased maximum intensity of LNa neuron activity to explore the role of the feedforward inhibitory neurons in the feeding state modulation (Fig. 3h-j). Another version of the model was designed with the sucrose state represented by adding an input to Handle-b neurons to explore the contribution of feedback inhibitory neurons (Fig. 3k-m). Finally, we constructed a combined model in which both LNa and Handle-b neurons were modulated in the sucrose-fed state (Fig. 3n-p, Supplementary Fig. 4a-f). In all versions of the model, the Hunching decreased and Head Casting increased in the sucrose-fed state compared with the normal fed state (Fig. 3j, m, p), which is in good agreement with the experimental results.”

3) In the new description of the supervised classifier, the statement "common actions may have very different features" is a bit confusing to me. Does this suggest that the same behavior looks different when in different feeding states? How, then, were the annotations performed? If they

were blinded, for example, this would seem to cause some issue if the same behavior looks different.

The behavior looks the same the a human annotators, but since some dynamics of the actions change (i.e. speed, amplitude), the features that were used for training the classifiers in the fed states may not cover the full range of values that are characteristics of these same actions in starved and sucrose-fed larvae, and/or larvae of different genetic backgrounds. Therefore new annotations (from these new experiments with different genotypes and in different states) were performed to retrain the classifiers.

We have updated the related manuscript in the text for clarification. It now reads:

"The new classifiers were trained on larvae fed on different diets, thus taking into account a broader range of behavioral dynamics and different phenotypes where common actions, e.g. such as Hunch, may have different features, e.g. they can exhibit slower or faster head retraction than Hunches detected previously."

Reviewer #4 (Remarks to the Author):
